# Marine Exopolysaccharide Complexed With Scandium Aimed as Theranostic Agents

**DOI:** 10.3390/molecules26041143

**Published:** 2021-02-20

**Authors:** Mattia Mazza, Cyrille Alliot, Corinne Sinquin, Sylvia Colliec-Jouault, Pascal E. Reiller, Sandrine Huclier-Markai

**Affiliations:** 1GIP ARRONAX, 1 rue Aronnax, CS 10112, F-44817 Nantes, CEDEX 3, France; mattia.mazza@subatech.in2p3.fr (M.M.); alliot@arronax-nantes.fr (C.A.); 2SUBATECH, 4 rue Alfred Kastler, BP 20722, F-44307 Nantes, CEDEX 3, France; 3INSERM U892-8 quai Moncousu, BP 70721, F-44007 Nantes, CEDEX 1, France; 4IFREMER, Institut français de recherche pour l’exploitation de la mer, rue de l’Ile d’Yeu, BP 21105, F-44311 Nantes, CEDEX 3, France; corinne.sinquin@ifremer.fr (C.S.); Sylvia.Colliec.Jouault@ifremer.fr (S.C.-J.); 5DES–Service d’Etudes Analytiques et de Réactivité des Surfaces (SEARS), Université Paris-Saclay, CEA, F-91191 Gif sur Yvette, CEDEX, France; Pascal.REILLER@cea.fr

**Keywords:** exopolysaccharides, scandium, theranostic, characterization, complexation

## Abstract

(1) Background: Exopolysaccharide (EPS) derivatives, produced by *Alteromonas infernus* bacterium, showed anti-metastatic properties. They may represent a new class of ligands to be combined with theranostic radionuclides, such as ^47^Sc/^44^Sc. The goal of this work was to investigate the feasibility of such coupling. (2) Methods: EPSs, as well as heparin used as a drug reference, were characterized in terms of molar mass and dispersity using Asymmetrical Flow Field-Flow Fractionation coupled to Multi-Angle Light Scattering (AF4-MALS). The intrinsic viscosity of EPSs at different ionic strengths were measured in order to establish the conformation. To determine the stability constants of Sc with EPS and heparin, a Free-ion selective radiotracer extraction (FISRE) method has been used. (3) Results: AF4-MALS showed that radical depolymerization produces monodisperse EPSs, suitable for therapeutic use. EPS conformation exhibited a lower hydrodynamic volume for the highest ionic strengths. The resulting random-coiled conformation could affect the complexation with metal for high concentration. The Log*K* of Sc-EPS complexes have been determined and showing that they are comparable to the Sc-Hep. (4) Conclusions: EPSs are very promising to be coupled with the theranostic pair of scandium for Nuclear Medicine.

## 1. Introduction

Osteosarcoma is the most common primary malignant bone tumor in children and young adults. It is a very aggressive type of cancer with a low prognosis, but molecular mechanisms of metastases have been identified for designing new therapeutic strategies in the frame of personalized medicine. Among the biomarkers considered, glycoaminoglycans (GAGs) are a hallmark related to cancer growth and spread. GAGs could be employed in new drug delivery systems to deliver active compounds for therapeutic in oncology [1]. GAGs such as unfractionated heparin, and its low-molecular weight (LMW) derivatives, are generally used for prevention or treatment of venous thromboembolism that frequently occurs in cancer patients. Heparin is composed of glucuronic or iduronic acid, glucosamine and has a sulfur content around 10% (from 8% to 13% according to the heparin preparation). An improved survival rate has been shown [2,3], which could be due to heparin, as it is able to interfere with some of the important biological steps in the metastasis spread [4]. Unfortunately, this kind of molecule exhibits drawbacks, such as hemorrhagic risk, and could present a contamination with prion proteins or oversulfated chondroitin sulfate [5]. This hazard profile drastically limits the use of heparin in therapy. Actually, the number of studies on the exploration of heparin mimetic molecules has been raised in recent years. Among the possible molecules, polysaccharides could be a new source of heparin-like molecules. Traditional algal or bacterial polysaccharides provide a valid alternative to mammalian GAGs [6]. Additionally, polysaccharides extracted from eukaryotes and prokaryotes living in marine environment could potentially reveal a better benefit/risk ratio as observed in preliminary animal studies [6]. In the past, algal polysaccharides such as fucoidans—and especially low-molecular-weight fucoidan preparation (LMWF)—have been shown to exhibit heparin-like properties with very low hemorrhagic risk [7]. Marine polysaccharides from a bacterial source also present a high potentiality in cell therapy and tissue engineering. They can be produced under totally controlled conditions in bioreactors [8] compared to other polysaccharides from eukaryotes.

Among these marine polysaccharides, an exopolysaccharide (EPS) of interest is extracted from the culture broth of a bacterium named *Alteromonas infernus*. The repeating unit of this EPS consists of a monosulfated nonasaccharide composed of six neutral hexoses and three hexuronic acids, among those the galacturonic acid unit bears one sulfate group. It presents a high molecular weight (>10^6^ g mol^−1^) and a low sulfate content (<10%) [9,10,11]. This native EPS demonstrated a very weak anticoagulant activity compared to heparin. In order to promote its biological activities and to provide a GAG mimetic compound, the native EPS has been modified by first a radical depolymerization, then followed by an oversulfation step [12,13]. EPS-DR represents the native EPS that has undergone radical depolymerization, while EPS-DRS has undergone additional over-sulfation. These two chemical modifications allow for a reduction of its molecular weight along with an increase in its sulfate content. The anticoagulant activity of this highly sulfated EPS derivative is improved compared to that of its native precursor but remains lower than both unfractionated heparin and LMW heparin, 10 and 5 times less, respectively. A hypothetical structure of the oversulfated exopolysaccharide (EPS-DRS) has been proposed [12,13] and is presented in Figure 1a. In the proposed structure three hexose units could be sulfated after the oversulfation reaction according to preliminary mass spectrometry analysis, as indicated in Figure 1a [12,13,14].

The EPS DRS could effectively inhibit, in vitro, both the migration and invasiveness of osteosarcoma cells [15]. It has been shown different levels of bone resorption regulation due to the presence of over sulfated-EPSs, most of them leading to proresorptive effects [13]. Moreover, this compound has been evidenced to be very efficient at inhibiting the establishment of lung metastases in vivo [15]. These EPSs could pave the way for the development of new drug-delivery systems in bone oncology in the field of personalized medicine [16].

In cancer treatment, personalized medicine uses specific information about a patient’s tumor to help making a diagnosis, planning treatment, finding out how well treatment is working, or making a prognosis. Examples of personalized medicine include using targeted therapies to treat specific types of cancer cells, such as HER2-positive breast cancer cells, or using tumor marker testing to help diagnosing cancer [17]. Nuclear medicine enables clinicians identifying non-invasively and precisely specific molecular activity within tissues and organs of the body, facilitating the early detection of disease and the immediate monitoring of therapeutic responses. Nuclear medicine involves the use of radioactive isotopes in the diagnosis and treatment of diseases. Theranostics—a field of medicine that combines specific targeted therapy based on specific targeted diagnostic tests—uses specific biological pathways in the human body, to acquire diagnostic images and to deliver a therapeutic dose of radiation to the patient. The acquired image allows for identifying the patient response and estimating, for the responder, the dose of the therapeutic agent to be injected. With this strategy, it is possible to give the right treatment, to the right patient, at the right time, with the right dose. For that purpose, it is necessary to have access to two specific radionuclides: one for imaging and another one for therapy. Such a pair of isotopes can be of a same element—often called true theranostic pairs, e.g., ^64^Cu/^67^Cu, ^44^Sc/^47^Sc, ^124^I/^131^I—or of element sharing, quite similar biodistributions—e.g., ^68^Ga/^177^Lu, ^99m^Tc/^188^Re [18,19]. Among the available radionuclides, scandium possesses two radionuclides emitting β^+^ radiation, ^44^Sc or ^43^Sc, which makes them appropriate candidates in PET/CT (Positron Emission Tomography—Computed Tomography) diagnosis, due to their half-life of around 4 h and decay to non-toxic Ca. For both radionuclides, half-lives are compatible with the pharmacokinetics of a fairly wide range of targeting vectors, such as peptides, antibodies, antibody fragments, and oligonucleotides. The validity, usefulness and advantages of ^44^Sc (E_β_ = 1 157 keV) have been demonstrated, e.g., featuring ^44^Sc-radiolabeled targeting vectors, including ^44^Sc-radiopharmaceuticals, very recently used in patients [20]. Some recent studies have shown also the validity of ^43^Sc-radiolabeled targeting vectors [21]. Finally, ^47^Sc is a low energy β^−^-emitter (T_1/2_ = 3.34 days; E_β_ = 270 keV; Eγ = 159.37 keV), which is suitable for SPECT (Single-Photon Emission-Computed Tomography) and planar imaging. ^47^Sc, which can be used for targeted radionuclide therapy, opens the field of Sc-based vectors from diagnosis to therapy and gives a great opportunity for dosimetric calculations. These exopolysaccharides, coupled with theranostic radionuclides, may serve as new radiopharmaceuticals drugs.

For use in a theranostic approach, the scandium-exopolysaccharide complex must be thermodynamically stable and kinetically inert. Hence, the aim of this work is to obtain these data, compared to heparin, considered as a reference. It is also important to characterize EPS –DR, -DRS and heparin as reference, especially their molecular masses. Asymmetrical Flow Field-Flow Fractionation (AF4) technique coupled to differential Refractive Index (dRI), and Multi Angle Light Scattering (MALS) detectors was used. Usually, Size-Exclusion Chromatography (SEC) coupled to MALS is used for the determination of weight-average molar mass ($\bar{M_{w}}$) of macromolecules [22,23]. However, AF4 has been recognized to be a suitable technique for the characterization of molar mass distribution of synthetic and natural polymers [24,25,26] as well as for nanoparticles [27], that are considered as drug delivery systems in antitumor and anti-metastasis therapy approaches [28].

Another important parameter that could influence the complexation is the polymer conformation [29]. To this aim, the hydrodynamic volume at different ionic strength was measured. Finally, to prevent the potential hydrolysis of EPS ligands, the stability constants Sc-EPS was assessed through Free Ion Selective Resin Extraction (FISRE) and the influence of sulfur content on the complexation with scandium was scrutinized. A good compromise has to be found between the complexes’ stability and the sulfur content, which is inversely linked to hemorrhagic risk.

## 2. Results

### 2.1. EPS ∂n/∂c

The ∂n/∂c values obtained were analyzed using the light scattering software ASTRA^®^ (see experimental section). The measured dRI values are then plotted against the concentration and fitted to a linear regression (Figure 2), the slope of which corresponds to the specific refractive index increment of the analyte in the given experimental conditions.

Usually, $\bar{M_{w}}$ values of the two EPSs are routinely determined by SEC-MALS, after the native EPS modifications. In the present paper, we have characterized the two EPSs through AF4 coupled to MALS. Table 1 summarizes the different values of ∂n/∂c determined for the two EPS samples. In addition to the detector constant and concentration (or injected molar mass), signal intensity also relies on a sample-related constant and may be influenced by the molar mass (*M^x^*). The molar mass exponent *x* is equal to 1 for a molar mass sensitive detector (such as MALS) and is 0 for a typical concentration detector (such as dRI). When using either a dRI or a MALS detector, the sample related constant is ∂n/∂c. As shown in Table 1, the ∂n/∂c of EPS-DR and -DRS appears lower than the range commonly considered for polysaccharides (0.140–0.150 mL g^−1^) [30]. The ∂n/∂c of homogeneous polymers is constant and can be considered as a contrast factor. Thus, the slightly lower ∂n/∂c value indicates a lower signal intensity than for the average range of polysaccharides. It is also important to note that, as the ∂n/∂c increases, the molar masses that can be detected is lower (under the same analytical conditions). The lower value of ∂n/∂c would probably lead to a lower accuracy and precision on the lowest molecular mass determination of EPS.

### 2.2. Determination of EPS Molar Masses and Dispersion

The most commonly used technique for the separation and molar mass determination of polymers is SEC, whereas AF4 is very useful for the separation of (bio)polymers [31]. It can be coupled to MALS for the determination of polymer molar masses [22,23,24], easily replacing SEC-MALS that has been routinely used to characterize polysaccharides. AF4 is the method of choice, especially for high-molar mass samples that could avoid shearing degradation of long-chain polymers, as in SEC. In AF4, the smallest polymer chains are eluted first due to their small sizes, whereas the bigger ones are eluted last.

The fractogram of EPS-DRS is shown in Figure 3. The light scattering peak looks uneven and sharper than those in UV-VIS and dRI, which is typical of the highly charged polysaccharides analyzed with AF4-MALS [32]. The small peak shift is due to the delay from one detector to another one due to the tubing, which was not corrected in ASTRA.

During light scattering calculations, it is assumed that the value of ∂n/∂c remains constant across the given sample separation. Measurements of scattered light at different detection angles θ (Debye plot) are thus necessary. The *R_θ_*/*K**c values are calculated from Equation (2), knowing the ∂n/∂c value and the Rayleigh ratio *R_θ_* for the different θ. ASTRA software allows the direct extrapolation of the molecular weight of EPS-DR and -DRS from the scattered light measured after the AF4 separation.

The fractogram in Figure 4a displays the molar mass of EPS-DR extrapolated from the Debye plot (Figure 4b, with the 95% hyperboles of uncertainties). The second peak of the fractogram in Figure 4a was only considered for the Debye plot, since the first one corresponds to the void peak. For the linear regression (r^2^ = 0.9875) of EPS-DR, the slope corresponds to *R_θ_*, while the intercept $\bar{M_{w}}$. P_(θ)_ allows for obtaining the $\bar{M_{w}}$ value. For the measurements made at low angle and infinite dilution, since *P*(0) = 1, the intercept provides the $\bar{M_{w}}$ = (26.1 ± 3.2) kDa. For EPS-DRS, the regression provides an $\bar{M_{w}}$ = (57 ± 8) kDa (r^2^ = 0.9714). The results of molar mass and dispersity determined by AF4-MALS for the two EPS are compared in Table 2. Moreover, the angular dependence scattered light at low angles can be related directly to the z-average of the “radius of gyration” (R_g_). This is the only technique that can be used to measure the dimensions of the molecules without assumptions about their shapes.

### 2.3. Polymer Conformation

The first step in the analysis of any samples is the instrument or detector calibration. The response of the detectors is calibrated to a known property by the use of a narrow standard. In our case, a water-soluble anionic polymer, such as polyacrylic acid (PAA) was used. Intrinsic viscosities [η] of PAA solutions of different molecular weights—i.e., 2000, 8000, 10,000, and 50,000 g mol^−1^—were measured. Mark–Houwink plots of log intrinsic viscosity as a function of log $\bar{M_{w}}$ were calculated for the PAA (Table 3). From Equation (10), a value of a = 0.617 was obtained for PAA in NaCl 0.1 mol L^−1^ at 20 °C, which would indicate that the solvent could be considered as a good solvent—a = 0.5 to 0.7 being expected for a random coil polymer in a good solvent. From Equation (9), *Mv* was calculated together with [η], to determine the hydrodynamic volume *V_h_* from Equation (5) (see Section 4.5) [33].

Since polysaccharides are polyelectrolytes, their viscosimetric behavior was studied as a function of the concentration in water and in dilute aqueous solutions containing variable amounts of NaCl. This is also of importance since, for polyelectrolytes, as with polysaccharides of interest, the second virial coefficient is very sensitive to ionic strength. The results of the relative viscosity ln *η_r_* of EPS-DR and heparin are presented in Figure 5. In Figure 5a,b,e, the relative viscosity increases with polyelectrolyte concentration, and is progressively suppressed with the increase in the salinity of the solvent. This was in agreement with the viscosimetric behavior of other types of polyelectrolytes [34]. The ln η_sp_/c value allows for estimating [η] according the following formula.
(1)$$\left[\eta \right]=\underset{C\to 0}{\lim}\left(\frac{\ln\left(\eta_{r}\right)}{C}\right)$$

When plotting ln η_sp_/c vs. c (Figure 5c,d,f), it is observed that, at low polyelectrolyte concentrations, ln η_r_/c increases with decreasing c. This effect is caused by the increase in the Debye length with dilution, which reduces the screening of the charges fixed on the chain and strengthens their mutual repulsion. This is in agreement with the observation of Pavlov et al. [35] on highly charged polyelectrolytes. As a result, the size of the polyion increases. The behavior of EPS-DR and -DRS observed at the highest ionic strengths (Figure 5e), could be explained by a more important folding of the molecules induced by their larger structure.

The decrease in [η] could be fitted by linear regression as followed: log [η] = 0.259 log [NaCl] + 0.732 (r² = 0.9856) for EPS-DR; log [η] = −0.193 log [NaCl] + 0.649 (r² = 0.9849) for EPS-DRS and log [η] = −0.168 log [NaCl] + 1.186 (r² = 0.9633) for heparin. Hence, log [η] is slightly higher for heparin than for EPS-DR or -DRS. The intrinsic viscosity provides information on the compaction of flexible-chain polymers on passing to the moderate polymer concentration range in solvents of various thermodynamic quality. This behavior in solution was confirmed through the determination of the hydrodynamic volume *V_h_* of EPS-DR, -DRS, and heparin, at background electrolyte concentrations (Figure 5f). The same logarithmic trend as for [η] was observed with y = −0.19 ln(x) − 42.67 (r² = 0.9856) for EPS-DR; y = −0.19 ln(x) − 45.18 (r² = 0.9849) for EPS-DRS and y = −0.26 ln(x) − 44.89 for heparin (r² = 0.9633). The rate of decrease in *V_h_* was comparable between both EPS but slightly higher than for heparin. The uncertainties on the slopes are similar for the 3 compounds, i.e., with 0.14. In addition, the *V_h_* value is decreasing with increasing ionic strength (Figure 5f). This result could be translated when dealing with the complexation with metal ions. With decreasing ionic strength of the solution, the chain size increases due to electrostatic repulsion between charges. This aspect would be taken into consideration when evaluating the stability of polymer complexes with Sc^3+^ in the following section. For weak polyelectrolytes, with further decreasing concentration, a maximum concentration dependence of ln η_sp_/c was observed, and Table 4 gives a summary of the hydrodynamic sizes from viscosimetric measurements compared to the R_g_ determined by AF4.

The viscosimetric radius Rη (~Rh) values are slightly lower that the R_g_. Both R_g_ and Rh can be used to gain insight into the third key area of polyelectrolyte characterization: structure. The characteristic R_g_/Rh value for a globular protein is ~0.775 [36], which means that R_g_ is smaller than Rh. However, when molecules deviate from globular to non-spherical or elongated structures, then R_g_/Rh tend to values upwards of 0.775, as R_g_ becomes larger than Rh. This is, thus, the case of the two EPS and heparin.

### 2.4. Stability Constants of Sc-EPS and Sc-Heparin Complexes

Free Ion Selective Radiotracer Extraction (FISRE) was used to obtain the apparent stability constants of Sc-EPS and -heparin complexes, as explained elsewhere [37]. Other techniques, such as potentiometric titrations and CE-ICP-MS were considered as being less suitable to analyze EPS-Sc complexes. Potentiometric titrations could affect the integrity of the polymer chain, by exposing it to very acidic and very alkaline pH hydrolysis [38,39], whereas CE-ICP-MS could not be used because of the highly charged polysaccharides-metal complexes, leading to a difficult interpretation of the speciation.

The partition coefficients normalized over the partition coefficient without organic ligands were calculated as a function of ligand concentrations (Figure 6).

As detailed in § 4.6, the number of different complexation sites is related to the number of plateaus in the isotherm, and the slope indicates the stoichiometry of sites implied in the scandium complexation with ligand. From Figure 6, for each different ligand, only one kind of complexation site was evidenced with a stoichiometry of 1:1 with regard to the free Sc^3+^ ion under our experimental conditions. Complexation constants could be calculated by the fitting of log*K*d_norm_ vs. ligand concentration. As studies as a function of pH cannot be performed without potential ligand hydrolysis, the determined apparent complexation constant cannot be corrected from acid-base properties of ligand and potential H^+^ implied in the reaction. Table 5 summarizes the apparent constants at pH 6.1 and the deduced ones, implying free aquo Sc^3+^.

## 3. Discussion

The potential of polysaccharides such as GAGs in medicine has already been assessed [5], especially in tumor targeting [15,40,41], but not their coupling with theranostic radionuclides. The major interest in studying materials containing metal complexes of polymers stems from the fact that the rheological and other material properties of polymeric systems are strongly affected by complexation with metals [42].

### 3.1. EPS Molar Mass Determination and Polydispersity

Polysaccharides are heterogeneous substances; their chemistry, morphology, and polydispersity play an important role on their biodistribution [34,43], and when coupled to radionuclides, it could lead to the irradiation of healthy tissues. After their production in bioreactors, it is important to monitor their rheological properties to ensure reproducibility. AF4, an alternative separative technique to SEC-MALS with a higher selectivity [44], was proposed to determine the molecular weights of polysaccharides. MALS provides a direct and absolute way of measuring the $\bar{M_{w}}$ value [45]. Indeed, for polysaccharides, non-ideality arising from co-exclusion and polyelectrolyte effects can be a serious problem and, if not corrected for, can lead to significant underestimates for $\bar{M_{w}}$. At the concentrations used in this work, the non-ideality effect could be neglected. The estimate for $\bar{M_{w}}$ could thus be considered within a few percent of the true or “ideal” $\bar{M_{w}}$. In order to use MALS data, the ∂n/∂c value for the studied polyelectrolytes must be accurately known.

The ∂n/∂c values determined for the two EPS samples were lower than the common range considered for other polysaccharides. The published ∂n/∂c values for various macromolecules [30], particularly for polysaccharides such as dextran and pullulan, indicated ∂n/∂c values in aqueous solution ranging between 0.14 and 0.15 mL g^−1^, whereas the range is broader, i.e., from 0.044 to 0.218 mL g^−1^, for non-aqueous systems [45]. For chitosan, the degree of substitution could influence the ∂n/∂c value, particularly if ionic functional groups are involved in the polysaccharide structure [46]. The presence of counter ions, such as Na^+^, together with a small chain branching, were shown to decrease the ∂n/∂c value from the general range of aqueous solutions, i.e., 0.14–0.15 mL g^−1^ [47]. The different ∂n/∂c values between the two EPSs of this work could be explained by: (i) the differences in the sulfate contents, thus leading to different degree of ionization; by (ii) the different chain lengths of the two EPSs, as described later.

The ∂n/∂c values determined for EPS-DR and -DRS were thus used with AF4-MALS, and the $\bar{M_{w}}$ were determined. EPS-DR and -DRS appeared to be monodisperse. Their $\bar{M_{w}}$ were slightly higher than the ones published on low molecular weight polysaccharides, i.e., 10^6^ and 24–35 10^3^ g mol^−1^, respectively [13]. Nonetheless, it was confirmed that radical depolymerization provided homogeneous polysaccharides in terms of molar mass dispersion (for both $\bar{M_{n}}$ and $\bar{M_{w}}$) [48]. Monodisperse polymers have the same degree of polymerization or relative molecular mass [49], with a polydispersity index PDI = 1. For this kind of polysaccharide, the PDI found is usually around 1.5 [14], while most of the natural polysaccharides have PDI above 2 [50,51].

It can be assumed that the $\bar{M_{w}}$ of EPS-DR and -DRS were underestimated in the previously published work [14] since the authors characterized these EPS by SEC-MALS with an average ∂n/∂c values for both EPS. Many commonly used SEC stationary phases may contribute to the unwanted interactions between charged analytes and column material. Another limitation of SEC is the incapability of the technique to characterize high-molar mass polymers and polysaccharides accurately (M_w_ > 1 × 10^6^ g mol^−1^), even if SEC has been successfully used for characterization of commercial cationic hydroxyethyl cellulose derivatives [52]. Additionally, it has been well established that the potential for degradation exists when analyzing long linear polymers by SEC [53].

Finally, the use of SEC in combination with MALS or the triple detector system, where a true size exclusion from the column is not necessary in order to obtain absolute $\bar{M_{w}}$, would still give erroneous results as the ∂n/∂c for the polymer in solution would change due to the change in repeating unit composition. It was shown, for instance that loss of sulfate groups would result in the polymer assuming a less extended and more flexible conformation due to lower electrostatic intramolecular repulsion [39]. A desulfated polysaccharide molecule would, therefore, effectively behave as a smaller molecule in solution compared to another molecule of the same degree of polymerization but unchanged degree of sulfation. To some extent, the impact of this factor on elution from a SEC column could be minimized by using a mobile phase with high ionic strength, which will shield the electrostatic charges, but would definitely change the size of the polymer in solution [39].

Finally, both EPS samples exhibit nanometric sizes for their gyration radius that correspond to one quantitative indicator of molecular conformation. This last one controls the efficacy in drug delivery [54]. With R_g_ ranging from 44 to 58 nm, these EPS are suitable with rapid in vivo circulation, and fast clearance through the kidneys [43].

### 3.2. Polymer Conformation

Other useful quantitative indicators of molecular conformation are the viscosimetric radius (Rη) or the hydrodynamic radius (Rh). The hydrodynamic radius (or equivalent sphere radius) is usually determined by dynamic light scattering. Thus, this work has examined the viscosimetric radius (Rη) and the conformation of the polymers.

The viscosity of a polymer in solution is related to the polymer conformation, that could be affected by the complexation with metal ions [28]. The change in conformation was monitored through the polymer’s intrinsic viscosity in solution, by varying the ionic strength of solvent for both EPS-DR and heparin. Figure 5a,b show an important decrease in the intrinsic viscosity for all concentrations considered with increasing ionic strength. These results were in agreement with results otherwise obtained on heparin [55]. A decrease in Debye length and a shield effect are caused by cations from the background electrolyte that decreased the electrostatic repulsion of the ionized groups, thus leading to a drop in viscosity [56].

Indeed, electrostatic interactions are very long-range (ca. 1000 nm) in pure water but quickly become screened in the presence of added salt (10 nm for 10^−3^ mol L^−1^ monovalent salt). This is why, during dilution in pure water, polyelectrolytes adopt a more elongated conformation under the effect of electrostatic repulsions and the length of the chain is proportional to the number of monomers. However, the chain is not fully stretched, and it has been showed that the concept of the electrostatic blob allows for the accounting of both long-range electrostatic interactions at the monomers. When the conformation is a highly stretched chain bearing a large number of charges, thus producing an important interoperable electrostatic potential with the counter-ions.

Intrinsic viscosity, [*η*] and the hydrodynamic volume *V_h_* are related through Equation (9), where M is given by the Mark–Houwink–Sakurada equation (Equation (10)).

A universal calibration was performed first with polyacrylic acid (PAA) of different molecular weights. In this method, a set of standards are still required, but they do not have to be exactly the same. The polymer samples used for such calibration should be as narrow in distribution as possible because the width of the distribution will influence the values of *K* and *a*. Since viscosity is a function of molecular size and not strictly molecular weight, the constants *K* and *a* apply only to a given polymer at a specified temperature and solvent. They should be limited to the molecular weight range for which they were determined. The polymer must be linear, unbranched and not crosslinked. As a result, the calibration curve generated gives information about the intrinsic viscosity, molecular weight, and elution volume of a sample. Since intrinsic viscosity is the basis for this method, a universally calibrated system can be used for any polymer sample. One advantage of this method is that, unlike conventional calibration, the series of standards used do not have to be the same. Thus, a separate calibration for each type of polymer is not necessary.

In an aqueous solution at neutral pH, PAA was under an anionic form composed only of carboxylic groups (pK_a_ = 4.54 [57]). The PAA standards used were all linear. The intrinsic viscosity of each standard was determined (Table 3) and from this a calibration curve was established allowing the determination of *K* and *a* [58]. The *a* = 0.617 obtained for PAA in 0.1 mol L^−1^ NaCl at 20 °C would indicate that the solvent is a “poor” solvent. However, NaCl has a great affinity for water and the addition of ions to aqueous solutions of polymers reduced the hydration of hydrophilic polymers [59]. The polymers are partially hydrated due to the great affinity of ions for hydration and, consequently, the polymer–polymer interactions are enhanced. This led to a low *a* value, as observed experimentally.

The constants *K* and *a* from Equation (10) were used to determine the viscosity average molar mass (*M_v_*), depending on the polymer type and solute–solvent interactions. For instance, Grubisic et al. [60] calculated the *V_h_* of different linear and branched polystyrene using the product [*η*] $\bar{M_{w}}$. Hamielec and Ouano [61] determined the *V_h_* of standard polystyrenes using the product [*η*]$\bar{{\;M}_{n}}$. The authors concluded that inaccurate *V_h_* values were obtained, when using $\bar{M_{n}}$ or $\bar{M_{w}}$ instead of *M_v_* [62].

For the two EPS samples, a decrease in *V_h_* was observed with increasing ionic strength. This decrease is slightly higher for EPS-DR and -DRS than for heparin, meaning that EPS-DR and -DRS are more sensitive to the ionic strength changes. This could be explained by considering the overall charges surrounding EPS-DR or -DRS chains that are more important compared to the heparin ones. EPS-DR and -DRS exhibit more ionized functional groups that could be shielded by the counter-ions from the background electrolyte, leading to the folding of the polymer chain. The decrease in intrinsic viscosity with increasing ionic strength reflects a change in conformation: the expansion of the polymer chain is clearly affected by the addition of ions that seem to make the ionized functional groups of the polymer inaccessible. Coil expansion, and thus the intrinsic viscosity, decreased with ion concentration. This means that, with a higher *V_h_*, the EPS polymers seem to be more elongated, exhibiting a larger portion of available ionized functional groups, i.e., carboxylate, available for the counter ion, i.e., Na^+^. By contrast, when *V_h_* is lower, the polymer is more compact and tightly coiled, reducing the portions of ionized functional moieties for the counter ions.

Smidsrød and Haug [38] noticed that the size of the polysaccharide molecule in solution at a certain ionic strength changed depending upon the degree of desulfation (random coil to rigid rod) since the sulfate groups determine the total charge and charge distribution on the polymer. Loss of sulfate groups resulted in the polymer assuming a less extended and more flexible conformation due to lower electrostatic intramolecular repulsion. A desulfated polysaccharide molecule would, therefore, effectively behave as a smaller molecule in solution compared to another molecule of the same degree of polymerization but unchanged degree of sulfation. This is observed here between EPS-DR and -DRS. Hence, we measured the ability of polyelectrolytes to respond to changes in salt concentration by altering their hydrodynamic volume in agreement with published data on other type of polysaccharides [38].

### 3.3. Stability Constants of Sc-EPS and Sc-Heparin

As described by Santo et al. [42], the simplest model of complexation used in the literature is the non-bonded model, in which the metal atom interacts with the ligands simply by van der Waals and Coulombic interactions. This model cannot take into account the number and spatial arrangement of coordinating bonds specific to the particular metal, as the ligands tend to close-pack around the metal atom. In the bonded model, the metal atom is covalently bonded to ligands, and this preserves the coordination number and complex geometry. This model is applicable to stable metal-ligand complexes but cannot be used in the systems where ligand dissociation and exchange may occur.

The FISRE method has been used for the determination of thermodynamic stability of metallo-radiopharmaceuticals containing Y^3+^ [63], or for Sc^3+^ complexes with several chelating ligands, i.e., EDTA, DTPA, NOTA, and DOTA [37]. Other techniques, such as potentiometric titrations and CE-ICP-MS were considered being less suitable to analyze Sc-EPS complexes. The influence of coordination on the structure and rheology of polymer solutions has been extensively studied in the literature [42]. As written earlier, potentiometric titrations would affect the integrity of the polymer chain because of an extreme pH range that can cause chain scission. CE-ICP-MS, instead, could be used because of the complex highly charged polysaccharides-metal complexes, leading to a very difficult interpretation of the speciation.

Hao et al. demonstrated using NMR and FTIR measurements that metal atom interacts with the carbonyl oxygen of the polyvinylpyrrolidone (PVP) polymer and, based on the data, hypothesized possible coordination with amine group [64]. For sugar-based moieties, it was evidenced that, in glucuronic, as well as galacturonic acid, the complexation with a trivalent ion, such as Eu^3+^, occurred between the carboxylic moiety, the ring oxygen and the C-4 hydroxyl group [65,66].

To the best of our knowledge, there are no data on the complexation of EPS or heparin with Sc^3+^. It was shown that, on a sulfate containing polyelectrolyte, the fixation of weakly acidic cations resembled the formation of outer-sphere complexes, in which the cation was separated from the anionic group by a solvent molecule [67]. The cation and anion were held together by electrostatic forces, and the smaller the radius of the solvated cation, the stronger the bond. The stability constant of heparin with Ca^2+^ is log K = 2.09 at pH = 7.2 and I = 0.15 M [67]. Feofanova et al. determined by potentiometric titration the stability constants of heparin with divalent d-transition series metals, i.e., Cu^2+^, Zn^2+^, Co^2+^, Ni^2+^ and Fe^3+^ [65]. They found that the monoligand complex MHep^2-^ was the predominant species in weakly acidic, neutral, and weakly basic pH ranges. Otherwise, it was also shown that the affinity of metal ions for donor groups of low basicity is expected to decrease with increasing ionic radius of the hydrated cation. [68]. For Sc^3+^, the ionic radius is smaller than the one of Ca^2+^ [69], so a higher complexation is expected.

With increasing pH, metal hydrolysis occurs. Water molecules are present in the inner coordination sphere of the metal that are progressively replaced by hydroxo-complexes, such as [M(OH)Hep]^3−^ and [M(OH)_2_Hep]^4−^, with increasing pH. For all the metal-heparin complexes identified, these authors showed that log β values increased when more hydroxo-species of the metal were complexed—see Feofanova et al. [65] for application.

From a thermodynamic point of view, and with no further indication, the two following equilibria cannot be discriminated.
Sc^3+^ + H_2_O + Hep^4−^ ⇄ Sc(OH)Hep^2−^ + H^+^
Sc^3+^ + HepH_2_^2−^ ⇄ ScHepH^−^ + H^+^.

For EPS, these assumptions are not straightforward. Indeed, even if the literature provided some data on the complexation of polysaccharides with metals, there are very few data on the determination of their stability constants [70]. Nonetheless, considering the nonasaccharide monomer that composes EPS-DR and -DRS (Figure 1a), there are from five (DR) to eight (DRS) negative charges from its glucuronic and galacturonic acid units. Fuks and Bünzli [71] determined, by spectrophotometry, the complexation constants of these two latter uronic acids with lanthanide trivalent ions. These authors found that the complexation of La(III) with glucuronic acid and galacturonic acid lead to two complexes with respective values of log*K*_1_ = 1.32 and log*K*_2_ = 3.86 and log*K*_1_ = 1.41 and log*K*_2_ = 3.94, without consideration of La(III) hydrolysis, which is less extensive than Sc(III) [72].

Thus, for our investigated systems, the model with more than one complex was chosen, as it would be more relevant for the experimental data considered. The stability constants found for EPS-DR and -DRS were higher than those from Fuks and Bünzli [71] on glucuronic and galacturonic acids, and this could suggest the presence of hydroxo-complexes of Sc for both of them. Indeed, the hydrolysis of Sc^3+^ is more extensive than La^3+^ since it occurs at pH 5 [71].

Comparing the stability constants between two EPS, Sc-EPS-DR complexes seem to be slightly more stable than the Sc-EPS-DRS ones. However, in terms of speciation, both ligands seem to quantitatively complex scandium in conditions near physiological ones. In terms of charge, EPS-DRS present three additional negative charges, related to the three sulfate groups generated by the oversulfation reaction. The presence of the sulfate groups was awaited to enhance the complexation. The reasons for this opposite trend, instead, could be due to a higher steric hindrance with DRS monomers caused by sulfates. At higher ionic strength, those groups are shielded, which makes polymer folding into a more contracted random coil. Carboxylate moieties of EPS-DRS may be less accessible for steric reasons than those of EPS-DR. Nonetheless, Sc-EPS complexes appear to be less stable than those between scandium and classic chelate ligands used in radiopharmaceuticals, such as DOTA and DTPA (*K*_ScDOTA_ = 30.79; *K*_ScDTPA_ = 27.43) [73]. From the values published in [73], we calculated the apparent complexation constant for DTPA and DOTA, respectively, with scandium at pH = 6.1. We found 16.93 and 16.98, respectively. Even if these values are much higher than those determined for heparin, EPS-DR and –DRS (i.e., 6.38; 7.07 and 5.62 from Table 5), the complexation of scandium with these ligands is total. Additionally, the Sc-EPS and Sc-heparin complexes appear comparable, meaning that the complexation behavior of those exopolysaccharides is quite similar to the traditional GAG heparin.

Regarding the Metal–Heparin complexes, a comparison could be made plotting the log β values vs. the inverse effective ionic radius 1/IR of the free metals in Shannon [69] in Figure 7. It can be seen that Sc^3+^ seems to effectively show stronger complexation than Ca^2+^, and even stronger complexation than smaller ions, such as Zn^2+^, Ni^2+^ or Co^2+^. It is comparable to Mn^2+^ and Cu^2+^, which is distorted by Jahn–Teller effect. It also shows a markedly smaller value that the trivalent Fe^3+^ complex. In effect, the comparison could be somewhat difficult as the formation of Fe(OH)Hep complex—and of Cu(OH)Hep to a somewhat lesser extent—is more extensive than the other first series d-transition elements [65]. Even the formation of Sc(OH)Hep seems to be lower than Fe(OH)Hep.

The two EPSs studied are strongly complexing Sc^3+^ and thus, they can be further envisaged as carriers for the theranostic pair ^44^Sc/^47^Sc.

## 4. Materials and Methods

### 4.1. Chemicals

The exopolysaccharides used in this work, namely EPS-DR and -DRS, were kindly provided by the Institut Francais de Recherche pour l’Exploitation de la MER (IFREMER) of Nantes as already described elsewhere [12]. The native EPS is produced from fermentation, and then undergoes radical depolymerization and over-sulfation [12]. The sulfate content of both polymers was determined by elemental analysis (Central Microanalysis Department of the CNRS, Gif-sur-Yvette, France). These two EPS differ from their sulfate content (SO_4_^2−^ %) that were 10 and 48% for EPS-DR and -DRS, respectively. Several batches were produced in order to have sufficient amounts of EPS for performing the study. Each batch has been fully characterized using the methods described in the present paper.

Heparin sodium salt (from porcine intestinal mucose, ≥180 USP units mg^−1^), scandium chloride hexahydrate (purity 99.9%), polyacrylic acid (of different $\bar{M_{w}}$, 50,000, 10,000 and 8000 g mol^−1^) and certified bovine serum albumin (BSA) ($\bar{M_{w}}$ = 67,000 g mol^−1^, Sigma Aldrich, France), hydrochloric acid, sodium hydroxide, sodium chloride, ammonium acetate, and sodium azide were of analytical grade and were provided by Sigma Aldrich France. All other chemicals were pure reagent grade and used as received unless otherwise specified. Milli-Q water (18 MΩ.cm, Millipore) was used in all reactions. Heparin sulfate content was analyzed, similar to EPS, and showed 30% of SO_4_^2−^.

### 4.2. Refractive Index Increment (∂n/∂c)

The index increment (∂n/∂c) parameter of both EPS samples and heparin was measured in order to better extrapolate the molar mass from light scattering. ∂n/∂c refers to the rate of change of the refractive index with the concentration of a solution for a sample at a given temperature, a given wavelength, and a given solvent [74]. The approach used in this article to determine ∂n/∂c is generally the most accurate one, involving the so-called offline-batch-mode dRI experiment [29]. Here, a series of solutions of increasing concentrations of polysaccharides in a certain solvent are injected directly into the dRI detector. Concentrations of EPS range from 0.25 up to 2 mg mL^−1^ in Milli-Q water. The choice of Milli-Q water as a solvent is derived from the fact of comparing those literature values measured in H_2_O. The dRI measurements were performed at 25 °C, using an Optilab rEX (Wyatt Technology, Germany) with *λ*_0_ = 660 nm, and analyzed using the corresponding ASTRA^®^ software (Wyatt Technology, Germany).

### 4.3. Asymmetrical Flow Field-Flow Fractionation (AF4) coupled to Multi Angle Light Scattering (MALS)

The principles and applications of the Field–Flow Fractionation methods are largely described in the literature [31,75,76]. The eluent is forced through a vacuum degasser from the HPLC chain (Shimadzu, France) and then through a 0.1 µm on-line filter (Millipore Ltd., Watford, U.K.) into the Eclipse 3 (Wyatt Technology, Germany) before entering the flow-FFF channel (Wyatt Technology, Germany). The dimensions of the channel are 291×56×59 mm. A polyether sulfone (PES) membrane (5 kDa MWCO) with a 350 µm mylar spacer is used. The sample is injected through a sampling loop of 20 µL into the inlet of the channel with a 0.2 mL min^−1^ flow. The channel outlet is connected to the detectors. The AF4 device is simultaneously coupled on-line with a UV-visible spectrophotometer (SPD-20A; Shimadzu; λ = 254 nm), a differential refractometer (Optilab rEX, Wyatt Technology, Germany), and a multi angle light scattering detector (MALS) 18 angles (DAWN HELEOS II, Wyatt Technology, Germany). The flow rates are controlled employing the Eclipse software (Wyatt Technologies, Germany). The elution flow was fixed at 1 mL/min. Focalization flow was 2 mL min^−1^ and the cross flow was set in gradient, from 2 to 0 mL min^−1^. The carrier solution chosen was MilliQ water with 0.1% in volume of Tween 20. Molecular weight-certified bovine serum albumin (BSA) of 67,000 g mol^−1^ was used for calibration of the AF4 channel with a concentration of 1 mg mL^−1^, which corresponds to 1.5 × 10^−5^ mol L^−1^. To recapitulate, BSA is a protein, which is forming mono, di, tri and tetramers. The calibration performed under gentle conditions with high resolution to get complete base-line separation of the BSA monomer and the corresponding aggregates. For the polysaccharides, the concentrations were fixed at 2 mg mL^−1^. The injection conditions are summarized in Table 6. The optimization of these different parameters is further discussed in this paper. Each sample was injected at least in triplicate. Data were analyzed using ASTRA software. ASTRA^®^ is a versatile software available for the characterization (molar mass, size, conformation, conjugation, etc.) of macromolecules and nanoparticles via light scattering.

The determination of the concentration c is usually given through the dRI detector, whereas the *K** value is determined through Equation (2):(2)$$K^{*}=\frac{4\pi^{2}n_{0}^{2}}{\lambda_{0}^{4}N_{A}}{\left(\partial n/\partial c\right)}^{2}$$
where *λ*_0_ is the wavelength of the incident radiation, *N_A_* the Avogadro’s number and *n*_0_ the refractive index of the solvent.

The mass concentration determined when preparing the sample and the recovery ratio for these experiments was determined in order to get the final concentration of the sample analyzed through AF4.

The variation of MALS scattered light is directly proportional to the weight-average molecular weight, $\bar{M_{w}}$, according to the Zimm formalism [77],
(3)$$\frac{K^{*}c}{R_{\theta }}=\frac{1}{M_{w}P_{\theta }}+2A_{2}c$$
where *K** is the optical constant of Equation (2), *c* the concentration of the compound; *R_θ_* the Rayleigh ratio; *A*_2_ the second virial coefficient; *P_θ_* the form factor.

*A*_2_ reflects solvent-solute interactions whereas *P_θ_* describes the variation of scattered light with scattering angle *θ*.

The Rayleigh ratio *R_θ_* is defined as:(4)$$R_{\theta }=\frac{I_{s}\left(\theta \right)}{I_{0}\left(\theta \right)}$$

This represents the ratio between the amount of light scattered by the analyte solution in excess, and the amount of light scattered by the solvent at varying scattering angle *θ.* When extrapolated to zero angle, and at infinite dilution, the relation between *R_θ_* and $\bar{M_{w}}$ leads to the following equation.
(5)$$\bar{M_{w}}=\frac{R_{\theta }}{K^{*}c}$$

According to Equations (2) and (3), the light scattered at the different angles *R_θ_* depends on the (∂n/∂c)^2^ value, whereas the molecular weight *M_w_* depends on (∂n/∂c)^−1^ and the form factor *P_θ_*/*M_w_* depends on (∂n/∂c).

In this work, Zimm’s formalism is the one used for extrapolating the physical characteristics of the polymers from light scattering data. According to this formalism, the expression of *P_θ_* is the following:(6)$$P_{\theta }=1-\frac{u^{2}}{3}$$
where *u* is
(7)$$u=\frac{4\pi }{\lambda }n_{0}R_{g}\sin\left(\frac{\theta }{2}\right)$$

*R_g_* represents the gyration radius of the polymer and *n*_0_ is the refractive index of the solvent.

### 4.4. pH Measurements

The pH values were measured with a combined Ag/AgCl electrode (Metrohm), which was calibrated before each experiment by titration of degassed HCl 0.1 M solution by freshly prepared free carbonate NaOH 0.1 M solution to maintain constant ionic strength equal to 0.1 M. The electrode was calibrated using OPIUM software and the following equation.
(8)$$E=E^{0}+S\times \log\left[H^{+}\right]$$
where the additive term *E*^0^ contains the standard potentials of the electrodes used and contributions of inert ions to the liquid junction potential, *S* corresponds to the Nernstian slope. No significant deviation was observed in very acidic acid solution and alkaline solution and no correction was taken into consideration.

### 4.5. Viscosity Analysis

To investigate the polymer shape, the intrinsic viscosity at different ionic strength of solvent was determined. Five batches of 2 mL solutions containing EPS-DR were prepared in NaCl solution at different concentrations (from 10^−3^ to 0.1 mol L^−1^) and in MilliQ water. Three different concentrations of polysaccharide, i.e., 4.5, 9, and 18 mg mL^−1^, were prepared for each ionic strength. The same kind of analysis was assessed on Na-heparin (Sigma Aldrich, $\bar{M_{w}}$ ≈ 18,000 g mol^−1^) to have a reference of viscosity behavior in the same ionic strength conditions. Viscosimetric measurements were realized using a digital microviscometer Lovis 2000 M^®^ (Anton Paar, France). A rolling-ball viscometer measures the rolling time of a steel ball through a glass capillary filled with transparent or opaque liquids according to Hoeppler’s falling ball principle. The temperature was set at 20 °C.

The hydrodynamic volume *V_h_* of the polymer, that represents the sum of the time-average of the molecular volume and the volume of the solvent molecules associated with it, is calculated using Equation (9) [78].
(9)$$V_{h}=\frac{4\pi \left[\eta \right]M_{v}}{\mu }$$
with *μ* = 10π *N_A_*, where *N_A_* is the Avogadro’s number, [η] and *M_v_* are the intrinsic viscosity and the viscosity-average molar mass, respectively, that are present in the Mark-Houwink-Sakurada (MHS) equation [79].
(10)$$\left[\eta \right]=KM_{v}^{a}$$

The calculation of MHS parameters is carried out by the graphic representation of the following equation:

ln[η] = ln*K* + *a* ln*Mυ*(11)
where *K* and *a* are M-H constants, depending upon the type of polymer, the solvent, and the temperature of viscometric determinations.

The exponent *a* is a function of polymer geometry varying from 0.5 to 2.0. Since MHS constants for EPS-DR, *K*, and *a* are not known, they have been estimated using the polyacrylic acid (PAA) of different $\bar{M_{w}}$, i.e., 2000; 8000; 10,000; 50,000 g mol^−1^. These constants can be determined experimentally by measuring the intrinsic viscosity of several polymer samples for which the molecular weight has been determined by an independent method, e.g., osmotic pressure or light scattering. Using the polymer standards of PAA, a plot of ln[*η*] vs. ln $\bar{M_{w}}$ usually gives a straight line, the slope of which is *a* and its intercept is ln *K* [80]. The M-H-S exponent bears the signature of a polymer chain’s three-dimensional configuration in the solvent environment: *a* values from 0.0−0.5 reflect a rigid sphere in an ideal solvent, also called the theta solvent; those from 0.5−0.8 have a random coil in a good solvent; those from 0.8−2.0 have a rigid or rod like configuration (stiff chain) [81].

### 4.6. Free-Ion Selective Radiotracer Extraction (FISRE)

FISRE was used to determine the stability constants of a complex M:L, at tracer level, using a cationic exchange resin Chelex 100, which bears iminodiacetic functional groups [82]. The chelating resin competes with the ligands for Sc^3+^ complexation. FISRE was applied in batch mode, varying the concentration of polymer ligand. It allowed for determining the association processes.

The adsorption of Sc^3+^ by iminodiacetate chelating groups ($\bar{X-H}$) can be described by the following exchange equation:(12)$$Sc^{3+}+3\;\bar{X-H}⇄\bar{X_{3}-Sc}+3H^{+}$$

Over-lined species are related to adsorbed species present on the resin. The partition coefficient *K_d_* could be expressed as follows:(13)$$K_{d}=\frac{{\left[\bar{Sc\left(III\right)}\right]}_{ads}}{{\left[Sc\left(III\right)\right]}_{sol}}=\frac{\left[\bar{X_{3}-Sc}\right]}{\left[Sc^{3+}\right]\alpha_{Sc\left(L,OH\right)}}=\frac{K_{ads}{\left[\bar{X-H}\right]}^{3}}{{\left[H^{+}\right]}^{3}\alpha_{Sc\left(L,OH\right)}}$$
where *K_ads_* is the equilibrium constant for Sc^3+^ bound to the resin (Equation (12)), α*_Sc(L,OH)_* is the complexation coefficient of Sc^3+^, which defines the ratio between the total aqueous scandium concentration [Sc(III)]_sol_ and the Sc^3+^-aqua ion concentration [Sc^3+^], as shown in Equation (14).
(14)$$\alpha_{Sc\left(L,OH\right)}=\frac{{\left[Sc\left(III\right)\right]}_{sol}}{\left[Sc^{3+}\right]}$$

When the exchange resin is used in large excess in comparison to the initial concentration of scandium, $\left[\bar{X-H}\right]$, the free concentration of ligand can be approximated to Ce, the exchange capacity of the resin. Moreover, at fixed pH. Kd (Equation (13)) can be simplified:(15)$$K_{d}=\frac{K_{ads}{\left[\bar{X-H}\right]}^{3}}{{\left[H^{+}\right]}^{3}\alpha_{Sc\left(L,OH\right)}}=\frac{C1}{\alpha_{Sc\left(L,OH\right)}}$$
where *C*1 is a constant. The partition coefficient also varies only with the Sc^3+^ complexation in aqueous medium. The complexation coefficient varies with pH, ligand concentration and other complexing agents: β_OH_,_i_ are the hydrolysis constants corresponding to the equilibrium [83],
(16)$$Sc^{3+}+iH_{2}O⇌Sc{\left(OH\right)}_{i}+i\;H^{+}$$

β_Cl_,_j_ are the chloride complexation constants corresponding to the following equilibrium [72].
(17)$$Sc^{3+}+jCl^{-}⇌Sc{\left(Cl\right)}_{j}{}^{3-j}$$

Used thermodynamic constants are summarized below in Table 7:

The behavior of scandium in different systems is also influenced by the presence in solution of heparin or EPS ligands (L), due to the formation of the associated complexes. If we consider the different polymers as aqueous molecules with different potential complexation sites, an equilibrium can be defined for each reaction site,
(18)$$Sc^{3+}+l\;L^{p-}+h\;H^{+}⇌ScH_{h}L_{l}{}^{3-l\times p+h}$$
with the associated complexation constant.
(19)$$\beta_{h,l}=\frac{\left[ScH_{i}L^{3-p+i}\right]}{\left[Sc^{3+}\right]\left[L^{p-}\right]{\left[H^{+}\right]}^{i}}$$

Considering all the reactions occurring in aqueous medium, the complexation coefficient $\alpha_{Sc\left(L,OH\right)}$ can be expressed as
(20)$$\alpha_{Sc\left(L,OH\right)}=\frac{\left[Sc^{3+}\right]+\sum_{i=0}^{4}\left[Sc{\left(OH\right)}_{i}^{3-i}\right]+\sum_{j=1}^{3}\left[Sc{\left(Cl\right)}_{j}^{3-j}\right]+\sum_{h,l}\left[ScH_{h}L_{l}\right]}{\left[Sc^{3+}\right]}$$
$$=1+\overset{4}{\underset{i=1}{\sum }}\frac{\beta_{i}}{{\left[H^{+}\right]}^{i}}+\overset{3}{\underset{j=1}{\sum }}\beta_{j}{\left[Cl^{-}\right]}^{j}+\underset{h,l}{\sum }\beta_{h,l}{\left[L\right]}^{l}{\left[H^{+}\right]}^{h}$$

As the experiments were performed at fixed pH and constant chloride concentration, the complexation coefficient varies only with the aqueous concentration of different sites on heparin or EPS ligands. This concentration cannot be considered equal to the initial concentration due to its complexation with scandium. The mass balance of these sites must be solved to determine the different complexation constant.

Complexation constants could be calculated by fitting the dependence of the distribution coefficient (*K_d_*) of Sc(III) between the resin and the supernatant on the total ligand concentration in the supernatant. At very low concentrations of heparin or EPS ligands, they do not significantly affect the scandium speciation and the partition coefficient is also constant.
(21)$$\log Kd\approx \log C1+\log\left(1+\overset{4}{\underset{i=1}{\sum }}\frac{\beta_{i}}{{\left[H^{+}\right]}^{i}}+\overset{3}{\underset{j=1}{\sum }}\beta_{j}{\left[Cl^{-}\right]}^{j}\right)$$

When heparin or EPS ligand concentrations increase, the complexation becomes major, it can be approximated to $\alpha_{Sc(L,OH)}=\underset{h,l}{\sum }\beta_{h,l}{\left[L\right]}^{l}{{[H}^{+}]}^{h}$, where the equilibrium ligand concentration is equal to the initial ligand concentration.
(22)$$\log Kd\approx \log C1+\log\left(\underset{h,l}{\sum }\beta_{h,l}{\left[L\right]}^{l}{\left[H^{+}\right]}^{h}\right)$$
log*Kd* vs. log[*L*] allows determining the slopes, which are related to the stoichiometric coefficient l. The x-intercept between these slopes and the plateau corresponding to the adsorption of scandium without ligand is related to −log*β_h,l_*.
(23)$$\log\beta_{h,l}{\left[H^{+}\right]}^{h}=-l\times \log\left[L\right]$$

Concerning the experimental conditions, 10 mg of exchange resin were added to a final volume of 5 mL. Solutions contained fixed 10^−4^ mol L^−1^ Sc^3+^ and varying ligand concentration from 10^−9^ to 10^−3^ mol L^−1^. The resulting suspensions were daily monitored and adjusted to pH = 6. The separation of the solid and liquid phases was done by sedimentation. Aliquots of the supernatant from 0.4 to 1 mL were taken for ICP-AES analysis (ICP spectometer iCAP 6000 of Thermo Scientific). For the ICP-AES, samples were prepared in nitric acid 1% and run through an auto sampler ASX-520 (Cetac). The calibration for Sc was done in the range 0–100 ppb, prepared by a dilution of a standard solution of 10.04 μg mL^−1^ (PlasmaCAL, SCP Science). Sc was detected at 255, 366, and 424 nm. The average LQM was 0.576 ppb.

## 5. Conclusions

Two exopolysaccharides, a slightly sulfated named EPS-DR and a highly sulfated named -DRS, exhibiting heparin-like properties, have been characterized in terms of molar mass distribution and conformation. Their respective ∂n/∂c values were determined for being used with AF4-MALS for reaching their $\bar{M_{w}}$ values. Both EPS were monodisperse and exhibited nanometric sizes, making them suitable as targeting chelates for their further use in Nuclear Medicine. We confirmed the ability of these polysaccharides to respond to changes in salt concentration by altering their hydrodynamic volume in agreement with a polyelectrolyte’s behavior. The rheological tools developed in this work, were successfully applied to monitor the repeatability of the bio-production of these polysaccharides. The complexation properties of these EPS with scandium indicated that Sc-EPS-DR and -DRS stability constants were higher than those on glucuronic and galacturonic acids, thus suggesting the presence of hydroxo-complexes of Sc for both of them. Sc-EPS complexes appear to be less stable (*K*_ScEPS-DR_ = 9.14; *K*_ScEPS-DRS_ = 7.69) than scandium complexes with DOTA and DTPA, classically used in radiopharmaceutical drugs. However, this will not consist in a limitation of further use of these EPS since they exhibit biological properties, and especially antiproliferative properties in cancer cells. Further work will thus scrutinize, through biological assays, the synergetic effect of Sc-EPS complexes on different cancer cell lines, compared to the EPS and Scandium alone. In parallel, the optimization of the EPS radiolabeling with ^44^Sc may provide an additional purification step. The radiolabeling must provide the highest yield in a time frame compatible with the half-life of the radionuclide, in order to deliver the maximum activity to the patient, according to the clinical need. If both biological assays and radiolabeling yields are encouraging, then small animal studies will be envisaged with both Positron Emission Tomography imaging and biodistributions.

## Figures and Tables

**Figure 1 molecules-26-01143-f001:**
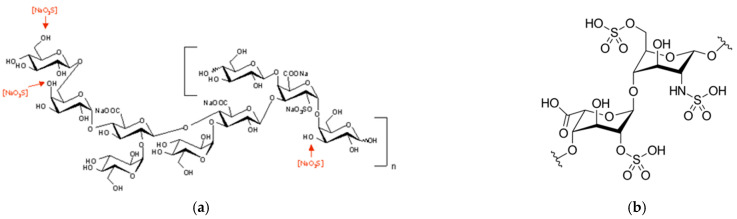
(**a**) Structure of EPS and postulated sulfated position after oversulfation reaction [12]. (**b**) Structure of heparin: the major repeating unit is the tri-sulfated disaccharide 2-O-sulfo-α-L-iduronic acid 1”4 linked to 6-O-sulfo-N-sulfo-α-D-glucosamine.

**Figure 2 molecules-26-01143-f002:**
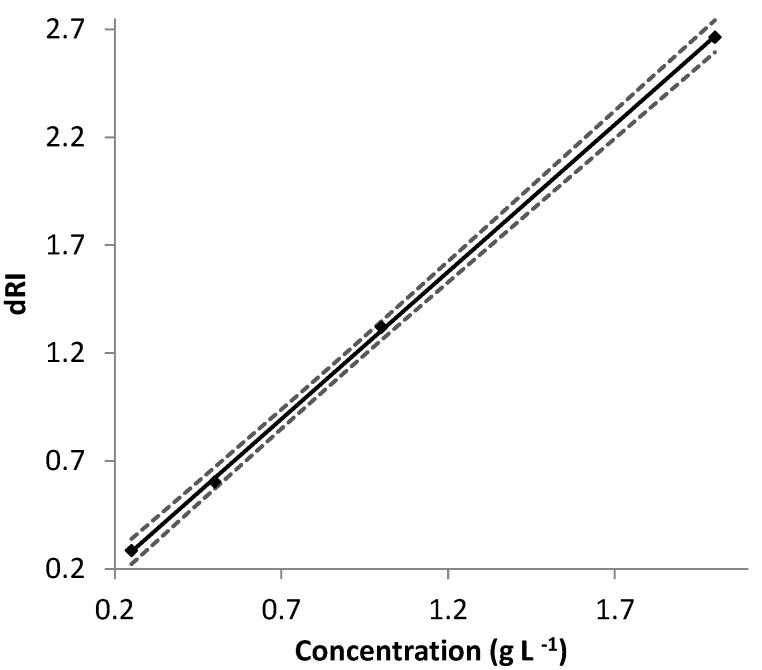
Determination of ∂n/∂c of EPS-DR in ultra-pure water at 25 °C; λ_0_ = 660 nm. Concentrations of EPS = 0.25 − 2 mg mL^−1^. Measured differential refractive index dRI values (× 10^4^) with EPS-DR concentration. The solid line represents the linear fit; the uncertainties’ hyperboles are plotted in dashed lines.

**Figure 3 molecules-26-01143-f003:**
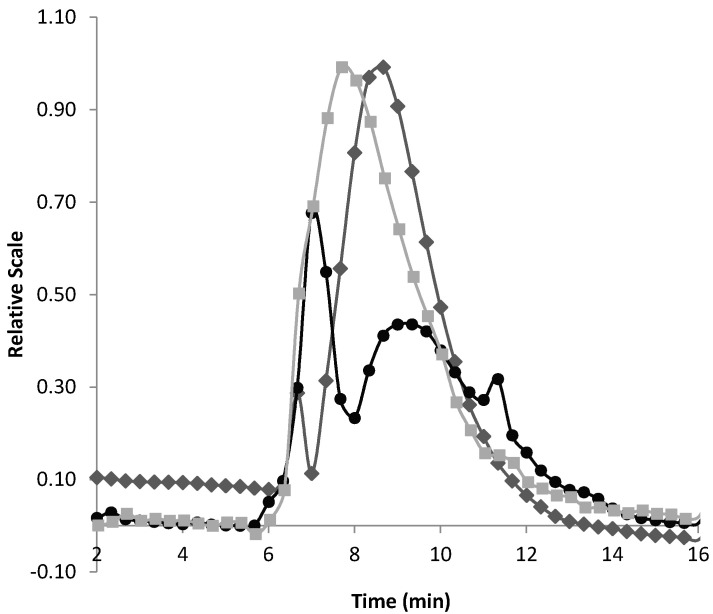
Fractogram of EPS-DRS (squares: UV-VIS @ λ = 254 nm; circles: dRI; diamonds:MALS @ 90°). [EPS-DRS] = 2 mg mL^−1^; injection volume = 20 µL; eluent = MilliQ water with 0.1% in volume of Tween 20; membrane = PES 5 kDa; canal length = 291 mm; spacer = 350 µm; cross flow = from 2 to 0 mL min^−1^ in 17 min; injection flow = 0.20 mL min^−1^; focalization flow = 2 mL min^−1^; detector flow = 1 mL min^−1^.

**Figure 4 molecules-26-01143-f004:**
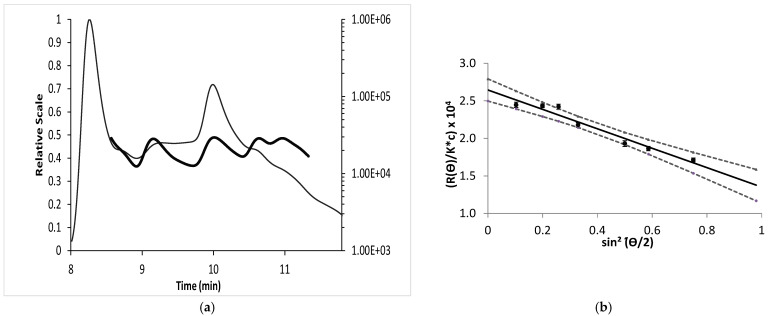
(**a**) Fractogram of molar mass of EPS-DR. The Light Scattering signal is plotted on the left Y axis, whereas the $\bar{M_{w}}$ calculated from this LS signal using ASTRA is plotted on the right Y axis. (**b**) Debye plot measurements of light scattering intensity at (θ) angles 18°, 36°, 45°, 58°, 90°, 108°, and 135° and analyzed with ASTRA.

**Figure 5 molecules-26-01143-f005:**
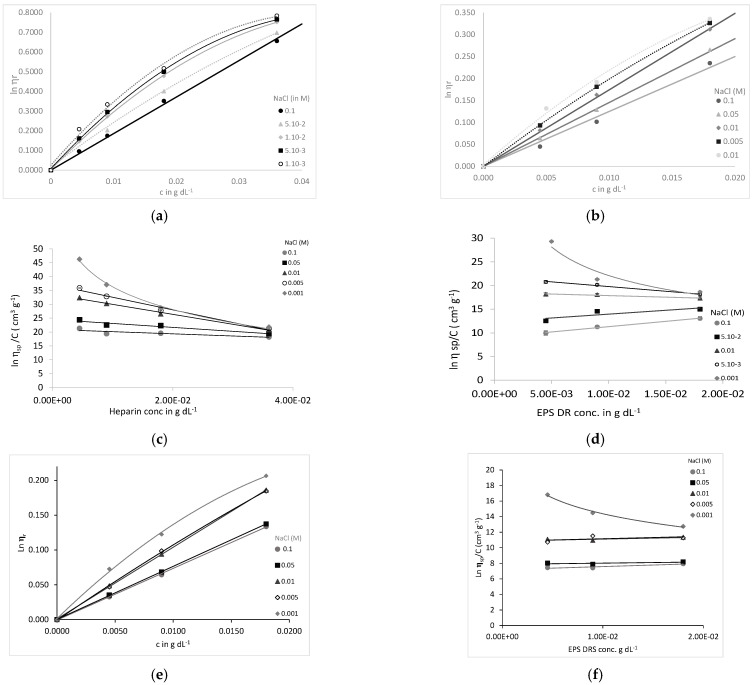
Relative viscosity η_r_ and intrinsic viscosity [η] at different ionic strength (NaCl) of (**a**,**c**) heparin, (**b**,**d**) EPS-DR and (**e**,**f**) EPS-DRS. Concentrations of EPS-DR, -DRS and heparin: 4.5, 9, and 18 mg mL^−1^ (additional concentration at 36 mg mL^−1^ for heparin). NaCl concentrations: 0.1 (circles), 5 × 10^−2^ (squares), 10^−2^ (triangles), 5 × 10^−3^ (empty diamonds), 10^−3^ mol L^−1^ (diamonds); (**g**) Intrinsic viscosity and (**h**) hydrodynamic volume of heparin (squares) and EPS-DR (circles) and -DRS (triangles) at different NaCl concentrations (in logarithmic scale).

**Figure 6 molecules-26-01143-f006:**
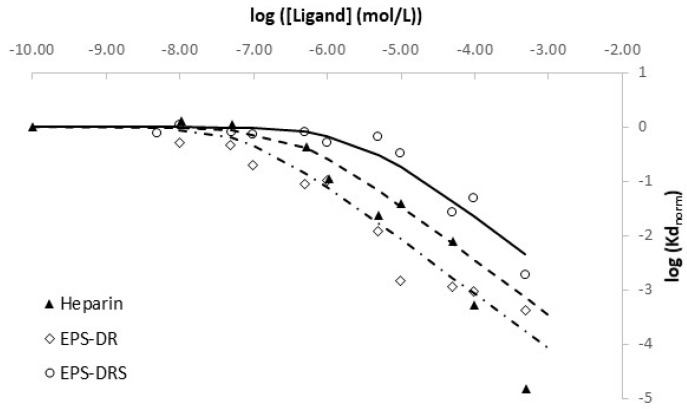
Ligand concentration isotherms of EPS-DR, EPS-DRS and Heparin: pH = 6.1 ± 0.1; 10 mg of chelex 100 resin in 5 mL of solutions; [Sc^3+^] = 10^−4^ mol L^−1^, [EPS-DR], [EPS-DRS] and [Hep] ranged from 10^−10^ to 5.10^−4^ mol L^−1^.

**Figure 7 molecules-26-01143-f007:**
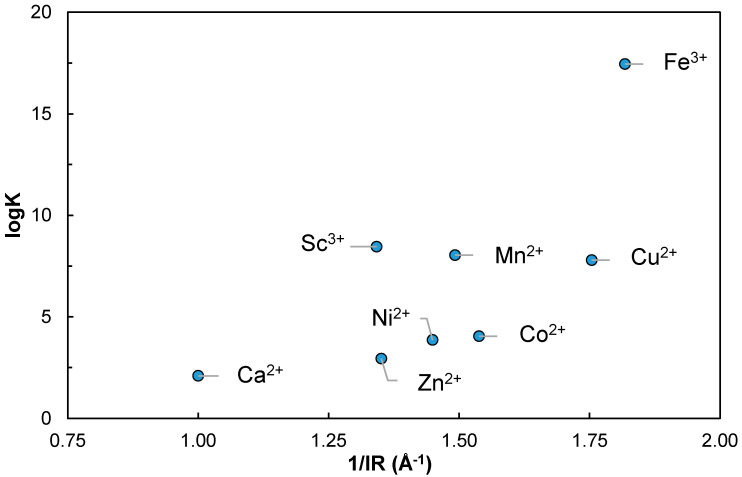
Comparison of log β for M-Heparin complexes from Feofanova et al. [65] (Zn^2+^, Cu^2+^, Ni^2+^, Co^2+^, and Fe^3+^), Rendleman [67] (Ca^2+^), and Sc-heparin in this work, as a function of the inverse effective ionic radius 1/IR [69].

**Table 1 molecules-26-01143-t001:** Determination of ∂n/∂c for EPS-DR and -DRS, and heparin in water at 25 °C and the molecular masses (uncertainties between brackets).

Sample	∂n/∂c (mL g^−1^)	SO_4_^2−^%	$\bar{M_{w}}\;(\mathrm{kDa})$
EPS-DR	0.125 (0.001)	9	26.6 (3.2)
EPS-DRS	0.110 (0.001)	48	57.0 (8.0)
Heparin	0.133 (0.001)	30	21.4 (2.2)

**Table 2 molecules-26-01143-t002:** AF4-MALS characterization in terms of molar mass and dispersity. The values in the brackets correspond to the uncertainties.

	$\bar{M_{w}}\;(\mathrm{kDa})$	$\bar{M_{n}}\;(\mathrm{kDa})$	$\mathrm{PDI}=\bar{M_{w}}$ $/\bar{M_{n}}$	R_g_ (nm)
DR	26.6 (3.2)	18.8 (2.8)	1.41 (0.38)	44.6 (0.1)
DRS	57.0 (8.0)	39.1 (4.1)	1.46 (0.36)	58.8 (0.1)

**Table 3 molecules-26-01143-t003:** Mark–Houwink calibration with PAA.

	PAA 2 kDa	PAA 8 kDa	PAA 10 kDa	PAA 50 kDa
Intrinsic Viscosity [η] (dL g^−1^)	3.03 × 10^−2^	5.83 × 10^−2^	10.32 × 10^−2^	21.51 × 10^−2^

**Table 4 molecules-26-01143-t004:** Size information summary. The values in the brackets correspond to the uncertainties.

	[η] ~ V_h_ (nm)	R_g_ (nm)	Mark Houwink	
				log K
PAA	/	/	0.617	2.785
Heparin	41.4 (0.3)	63.6 (0.1)		
EPS-DR	42.9 (0.3)	44.6 (0.1)		
EPS-DRS	43.8 (0.3)	58.8 (0.1)		

**Table 5 molecules-26-01143-t005:** Complexation constants obtained by FISRE for different ligand. Charges are omitted for the sake of simplicity.

	EPS-DR	EPS-DRS	Heparin
Apparent constant	7.07 ± 0.12	5.62 ± 0.12	6.38 ± 0.46
$\log\beta_{h,l}{\left[H^{+}\right]}^{h}$ (Equation (23))	9.30 ± 0.30	7.85 ± 0.30	8.61 ± 0.46

**Table 6 molecules-26-01143-t006:** AF4 method used to characterize EPS.

Start Time (min)	Duration (min)	Mode	Initial Cross Flow (mL/min)	Final Cross Flow (mL/min)
0.00	1.00	Elution	0.00	1.50
1.00	1.00	Focus		
2.00	2.00	Focus + Inject		
4.00	2.00	Focus		
6.00	10.00	Elution	1.50	1.50
16.00	7.00	Elution + Inject	1.50	0.00

**Table 7 molecules-26-01143-t007:** Thermodynamic constants of scandium(III) complexation.

Reactions	Value	Ref
Sc^3+^ + H_2_O ⇌ Sc(OH)^2+^ + H^+^	−4.16 ± 0.05	[83]
Sc^3+^ + 2H_2_O ⇌ Sc(OH)_2_^+^ + 2H^+^	−9.71 ± 0.30	[83]
Sc^3+^ + 3H_2_O ⇌ Sc(OH)_3_ + 3H^+^	−16.08 ± 0.30	[83]
Sc^3+^ + 4H_2_O ⇌ Sc(OH)_4_^−^ + 4H^+^	−26.7 ± 0.3	[72]
Sc^3+^ + Cl^−^ ⇌ ScCl^2+^	1.1 ± 0.1	[72]
Sc^3+^ + 2Cl^−^ ⇌ ScCl_2_^+^	0.75 ± 0.10	[72]

## Data Availability

Data available in a publicly accessible repository. The data presented in this study are openly available at [10.3390/molecules26041143].

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
