# Peer review of "Marine Exopolysaccharide Complexed With Scandium Aimed as Theranostic Agents"

_molecules, 2021, doi:10.3390/molecules26041143_

Round 1

Reviewer 1 Report

The manuscript entitled “Marine exopolysaccharide complexed with scandium aimed as theranostic agents” describes the determination of the mass (by AF4-MALS) and the conformation (by viscosity measurements) of two kinds of polysaccharides i.e. EPS DR and EPS DRS. Additionally, the determination of the stability constants of complexes of EPS DR and DRS with (non radioactive) scandium were carried out using a FISRE-based technique.

Additionally to the major and minor corrections listed below, to be accepted for publishing, the manuscript needs extensive polishing and a certain degree of re-structuring.

Major:

  1. The title promises scandium theranostic agents while the manuscript describes very little work done with non-radioactive scandium and no work at all done with radioactive scandium. The authors should either adapt the manuscript content to the title (i.e. adding more scandium data) or the title to the manuscript content (i.e. changing the title).
  2. All sections of the manuscript are too wordy. They should be informative but focused on essential findings and experimental data. The “Discussion” looks more like a lecture than a manuscript section: it contains unnecessary explanations that could be either removed or summarised in few essential words or even substituted by references (e.g. page 11 lines 337 to 347 and lines 377 to 386; page 12 lines 390 to 393 and 416 to 423 and 434 to 446). Some parts are disconnected (e.g there is no link between the first sentence in Page 1 line 31 and the following paragraph; Page 1 lines 37 to 39) and some sentences are confused (e.g. page 11 lines 366-368 and page 12 lines 422-424 where the verb is also missing; page 2 lines 78, 81, the authors talk about the possible use of EPSs for delivery systems, cancer therapy, determination of response to therapy, theranostics, nuclear medicine..some of that is off topic). The “Conclusion” is more a summary than a conclusion section suitable for research article. It is repetitive (one part is almost identical to a section of the Discussion) and too long. The authors should use this section to shortly point out the major achievements and indicate future plans. Additionally, there is a long series of imperfection throughout the whole manuscript (see minor corrections) which indicate lack of attention to details. Unfortunately, all the points mentioned above affect the fluidity of the manuscript.
  3. Page 6 line 221 and figure 4: The authors report Mw values for both EPS DR and DRS but Figure 4 shows only EPS-DR. Also Figure 7 shows results for EPS DR only. Can the authors give a good reason for not showing those data? If not, for clarity and completion, can the authors add the graphs/results related to EPS-DRS?
  4. It might be mentioned here and there in the manuscript, but can the authors describe in a straight-to-the point and clearer way the kind of EPS-Sc3+ complex they expect is formed (i.e. What might the coordination sphere be? What donor atoms are involved? Will there be water molecules in the complex?)? If possible, a figure would be useful.
  5. The stabilities of complexes of molecules with a non-radioactive metal can be (often) very different from the stability measured when the correspondent radioactive metal is used. Moreover, very often, it is not possible to prepare complexes with a radioactive metal (due to its very little amount available in solution) although the non-radioactive correspondent was successfully achieved. Based on that, the stability data described by the authors in the manuscript would be useless if the EPS-44Sc complex could not be achieved. If the authors could successfully prepare 44Sc-EPS complexes, would stability data using the radioactive compounds themselves be more realistic and reliable than the non-radioactive ones?

Minor:

  1. Although not the authors’ fault, there are a lot of formatting errors (e.g. page 3 line 105, 108; page 6 line 219; equation 11; page 8 line 264, 270, 280; and the t-rex (a favourite) in page 17 line 652) that forced a lot of guessing from the reviewer’s part.
  2. Page 2 line 61-62: Can the authors add a reference for the radical depolymerisation and oversulfation?
  3. Page 2 line 61: The authors use the same acronym for both the products and the processes to obtain them (i.e. EPS DR and EPS DRS). It might create some confusion. Would it be possible to differentiate the two things?
  4. Page 2 figure 1: Na3OSO- groups on the molecules: it looks like there are 3 sodium atoms (Na3), not three oxygen atoms (OSO3).
  5. Page 3 line 104: Scandium-44, with a half-life of 4h, is not compatible with the biological half-life of (full length) antibodies. The radiometal is more suitable for small proteins such as nanobodies and affibody molecules.
  6. The authors should define the following acronyms/abbreviations: page 3 line 108, T1/2; Page 3 line 121, SEC; Page 9 line 298, CE-ICP-MS; Page 21, line 757: ICP-AES; Page 21, line 762: LQM.
  7. Page 3, line 117: For clarity, in this sentence can the authors indicate specifically the name of the molecules they are talking about instead of calling them “these molecules”?
  8. Page 4, section from line 137 to line 163: This section might be more suitable for “Methods” than “Results”. And can it be stripped to its essential information?
  9. Page 5, line 181: the unit is missing
  10. Lack of consistency: table 1 the unit is mL/g and in the text is mL g-1; page 6 lines 214 and 222, AF4 is sometimes A4F; Figure 7, the graphs have different resolutions (some of them are difficult to read), inconsistent x-axis labels and legend/insert (a mix of decimal and base 10 numbers).
  11. Page 6, Figure 3: Is there a reason why the authors show the fractogram of EPS-DRS only and not of EPS-DR?
  12. Page 6, line 212: Are the authors sure of using equation 3 to calculate Rθ/K*c values? Page 7 line 242: Again, the authors refer to the wrong equation. Page 7, line 243: “see section 4.6” should be “see section 4.5”; Page 12, line 428: It should be “a”, not “α”.
  13. There is a little bit of inconsistency with the figures: figures 5 and 6 are missing. Page 12, line 410: the authors cite a non-existing figure 5.
  14. Page 8, line 273: “Higher than for heparin” not “higher for heparin”. The two sentences have very different meanings.
  15. Page 11, line 375: The authors say: “Another limitation of SEC is the incapability of the technique to characterize high-molar mass polymers”. What mass is considered high? Can the authors indicate some values in the text?
  16. Page 13, line 468: The authors say: “This could be explained by considering the overall charges surrounding EPS-DR or -DRS chains that are more important compared to the heparin ones”. What do the authors mean with the term “important” referred to a charge? Do the authors think that this sentence is essential for the story?
  17. Page 14, line 521 to 537 and Page 15, line 573-581: Why do the authors focus so much on heparin? In the current state, those paragraphs look off-topic. If important for the story, can the authors find a way to connect them to the main subject (i.e. EPS)?
  18. Page 17, section 4.3: The authors should fix some of the verb tenses. Line 641: what kind of filter was used (size, membrane, pore size)? what supplier?
  19. Page 21, lines 787 to 789: The authors talk about using the tools described in the manuscript to determine the consistency between different batches of polysaccharides. It is not clear if that was performed for this work or separately? Why not including those data in the manuscript?

Author Response

Reviewer #1:

The manuscript entitled “Marine exopolysaccharide complexed with scandium aimed as theranostic agents” describes the determination of the mass (by AF4-MALS) and the conformation (by viscosity measurements) of two kinds of polysaccharides i.e. EPS DR and EPS DRS. Additionally, the determination of the stability constants of complexes of EPS DR and DRS with (non radioactive) scandium were carried out using a FISRE-based technique.

Additionally to the major and minor corrections listed below, to be accepted for publishing, the manuscript needs extensive polishing and a certain degree of re-structuring.

Major:

  1. The title promises scandium theranostic agents while the manuscript describes very little work done with non-radioactive scandium and no work at all done with radioactive scandium. The authors should either adapt the manuscript content to the title (i.e. adding more scandium data) or the title to the manuscript content (i.e. changing the title).

The word “theranostic” not only refers to the use of radioactive compounds. According to Ballinger (Brit. J. Radiol. 2018 Nov; 91(1091): 20170969), the definition of theranostic involves the administration of a diagnostic agent:

  • to determine localisation in the site or disease state under study as a surrogate for a potential therapeutic agent with similar chemical properties;
  • to examine its biodistribution as predictive of off-target (adverse) effects of the potential therapeutic agent;
  • as an aid in determining the optimal therapeutic dosage or activity to be administered, based on the anticipated tumoricidal doses measured in the tumour site (e. dosimetry); and/or
  • to monitor the response to this treatment

The term “theranostic” is relatively recent, and also applies to therapies which do not involve systematically radionuclides. The concept goes back to the earliest days of nuclear medicine and nowadays, it is an active area of radiopharmaceutical research.

Additionally, thermodynamic stability and kinetic inertness are the most essential aspect to be considered when evaluating a chelator-radionuclide interaction in the development of a radiopharmaceutical. Each metal ion exhibits different physical and chemical properties, resulting in different coordination chemistries and therefore differing requirements on the chelator used. For establishing these chelation properties, the thermodynamic stability etc, it is not mandatory to work with radionuclides when the stable element exist, in accordance with the ALARA concept, and neglecting isotopic fractionation. This is what has been done in this work.

We agree with the reviewer’s comment on the fact that no work has been done with radioactive scandium in this work. If the associate editor agrees we would like to keep the title unchanged since it is clearly indicated “aimed as theranostic agent”.

  1. All sections of the manuscript are too wordy. They should be informative but focused on essential findings and experimental data. The “Discussion” looks more like a lecture than a manuscript section: it contains unnecessary explanations that could be either removed or summarised in few essential words or even substituted by references (e.g. page 11 lines 337 to 347 and lines 377 to 386, page 12 lines 390 to 393 and 416 to 423 and 434 to 446.

These sections have been reduced accordingly to the reviewer’s comment.

Some parts are disconnected (e.g there is no link between the first sentence in Page 1 line 31 and the following paragraph; Page 1 lines 37 to 39)

This part has been rephrased, as “Osteosarcoma is the most common primary malignant bone tumor in children and young adults. It is a very aggressive type of cancer with a low prognosis; but molecular mechanisms of metastases have been identified for designing new therapeutic strategies in the frame of personalized medicine. Among the biomarkers considered, glycoaminoglycans (GAGs) are a hallmark related with cancer growth and spread. GAGs could be employed in new drug delivery systems to deliver active compounds for therapeutic in oncology [1].”

and some sentences are confused (e.g. page 11 lines 366-368 and page 12 lines 422-424 where the verb is also missing; page 2 lines 78, 81, the authors talk about the possible use of EPSs for delivery systems, cancer therapy, determination of response to therapy, theranostics, nuclear medicine..some of that is off topic).

The different sentences that were confusing have been corrected. We do hope they are clearer.

page 11 lines 366-368 : The different ¶n/¶c values between the two EPS of this work could be explained by: (i) the differences in the sulfate contents, leading thus to different degree of ionization; and by (ii) the different chain lengths of the two EPS, as described later.

page 12 lines 422-424: A decrease of Debye length and a shield effect are caused by cations from the background electrolyte that decreased the electrostatic repulsion of the ionized groups, leading thus to a drop in viscosity [56].

page 2 lines 78, 81 the authors talk about the possible use of EPSs for delivery systems, cancer therapy, determination of response to therapy, theranostics, nuclear medicine..some of that is off topic)

We have rephrased the sentence “These EPS could pave the way for the development of new drug-delivery systems in bone oncology in the field of personalized medicine [16].”

We are sorry to disagree with the reviewer’s comment since the EPS will serve as chelates for being coupled with metallic radionuclides and will target the cells. The resulting system will also be used as a diagnosis companion, allowing a dosimetric assessment for further use in therapy in nuclear medicine.

The “Conclusion” is more a summary than a conclusion section suitable for research article. It is repetitive (one part is almost identical to a section of the Discussion) and too long. The authors should use this section to shortly point out the major achievements and indicate future plans.

We thank the reviewer for this comment. We agree with the fact that the conclusion was too long and without major prospective. We have thus differently written the entire conclusion.

“Two exopolysaccharides, a slightly sulfated named EPS-DR and a highly sulfated named -DRS, exhibiting heparin-like properties, have been characterized in terms of molar mass distribution and conformation. Their respective ¶n/¶c values were determined for being used with AF4-MALS for reaching their  values. Both EPS were monodisperse and exhibited nanometric sizes, making them suitable as targeting chelates for their further use in Nuclear Medicine. We confirmed the ability of these polysaccharides to respond to changes in salt concentration by altering their hydrodynamic volume in agreement with a polyelectrolytes behavior. The rheological tools developed in this work, were successfully applied to monitor the repeatability of the bio-production of these polysaccharides. The complexation properties of these EPS with scandium indicated that Sc-EPS-DR and -DRS stability constants were higher than the ones on glucuronic and galacturonic acids, and suggesting thus the presence of hydroxo-complexes of Sc for both of them. Sc-EPS complexes appear to be less stable (KScEPS-DR = 9.14; KScEPS-DRS = 7.69) than scandium complexes with DOTA and DTPA, classically used in radiopharmaceutical drugs. But this will not consist in a limitation of further use of these EPS since they exhibit biological properties, and especially antiproliferative properties in cancer cells. Further work will thus scrutinize, through biological assays, the synergetic effect of Sc-EPS complexes on different cancer cell lines, compared to the EPS alone and Scandium alone. In parallel, the optimization of the EPS radiolabeling with 44Sc, with may be an additional purification step will be done, The radiolabeling must provide the highest yield in a time frame compatible with the half-life of the radionuclide, in order to deliver the maximum activity to the patient accordingly to the clinical need aimed. If both biological assays and radiolabeling yields are encouraging then small animal studies will be envisaged with both Positron Emission Tomography imaging and biodistributions.”

Additionally, there is a long series of imperfection throughout the whole manuscript (see minor corrections) which indicate lack of attention to details. Unfortunately, all the points mentioned above affect the fluidity of the manuscript.

  1. Page 6 line 221 and figure 4: The authors report Mw values for both EPS DR and DRS but Figure 4 shows only EPS-DR. Also Figure 7 shows results for EPS DR only. Can the authors give a good reason for not showing those data? If not, for clarity and completion, can the authors add the graphs/results related to EPS-DRS?

The reviewer is right. Figure 4 shows EPS-DR fractogram but Figure 3 shows EPS-DRS fractogram. From these fractograms, the Debye plot has been established for both EPS in order to be able to determine their respective Mw. Here is the Debye’s plot of EPS-DRS. Since it is not bringing much information, and since only the extrapolation is allowing the determination of Mw, we have decided not to add this figure to the manuscript. Nonetheless, the linear regression for both EPS-DR and -DRS are indicated in the text.

Debye plot measurements of light scattering intensity at (θ) angles 18°, 36°, 45°, 58°, 90°, 108°, and 135° and analyzed with ASTRA for EPS-DRS.

For Figure 7, we have added the corresponding Figures for EPS-DRS. They are now figure 7 e and f.

  1. It might be mentioned here and there in the manuscript, but can the authors describe in a straight-to-the point and clearer way the kind of EPS-Sc3+ complex they expect is formed (i.e. What might the coordination sphere be? What donor atoms are involved? Will there be water molecules in the complex?)? If possible, a figure would be useful.

The free ion form Sc3+ is stable in acidic conditions below pH 4. For 4 < pH < 5.5, Sc undergoes to hydrolysis forming [Sc(OH)]2+ (logb°1= -4.16 ± 0.05) followed by hydrolysis in the form [Sc(OH)2]+ (logb°2 = -9.71 ± 0.30) at pH range 5.5-6.5. This latter further hydrolyses to Sc(OH)3 (logb°3 = -16.8 ± 0.30) at pH range 7-10. The Sc3+ free ion coordinates with electron-rich ligands: oxygen and nitrogen donors like hydroxyl, carboxylate, phosphate, and sulphate groups. Scandium forms with these ligands complexes in which the metal has a coordination number that is greater than 6 and up to 9, although the octahedral geometry is the most common. The stoichiometry of EPS-Sc3+ complex has been determined to be 1:1.

Without EXAFS or crystal structure determination, it would be quite speculative to propose a structure. No such attempt was made since it is not very relevant for Nuclear Medicine applications and outside of the objective of this work. The most important to this aim is the determination of the stability constant taking into account the hydroxo-species as explained above. This has been explained in the manuscript.

  1. The stabilities of complexes of molecules with a non-radioactive metal can be (often) very different from the stability measured when the correspondent radioactive metal is used. Moreover, very often, it is not possible to prepare complexes with a radioactive metal (due to its very little amount available in solution) although the non-radioactive correspondent was successfully achieved. Based on that, the stability data described by the authors in the manuscript would be useless if the EPS-44Sc complex could not be achieved. If the authors could successfully prepare 44Sc-EPS complexes, would stability data using the radioactive compounds themselves be more realistic and reliable than the non-radioactive ones?

We think that the reviewer is getting confused with the wording: i.e. (thermodynamic) stability constant vs. radiolytic stability. The former refers to the mass action law and the latter to the stability of a radiolabeled complex within ionizing radiation, which could lead to a demetallation of the complexes and the potential degradation of organic molecules. The thermodynamic stability constant does not depend on the metal concentration: or only marginally in the case of mass-dependent isotopic fractionation. the two isotopes have the same number of electrons, thus they both undergo the same chemical behavior.

By stability constant, it is understood that it is a conditional complexation constant (i.e. a mass action constant for the given experimental conditions that has not been extrapolated to standard condition as a thermodynamic constant). The reason why it has not been extrapolated to zero ionic strength is that it is quite impossible to determine exactly the ionic strength induced by the EPS themselves.

The complexation with the radionuclide is called radiolabeling. Since we have evidenced that Sc-EPS complexes exhibit a quite high complexation constant, radiolabeling will work. Labeling protocols should allow high labeling yields, high radiochemical purity and high specific activity. The challenge in producing radionuclides for nuclear medicine is to have carrier-free batches, i.e. without any stable element. The production of 44m/44Sc is already established, and allows reaching high purities batches (44Sc is carrier-free). Our group has published extensively on that (the most recent Cancer Biother. Radio. 2018, 33(8), 1-14; DOI:10.1089/cbr.2018.2485). Labelling efficiency of ligands is usually tested at different solution pH values, temperatures, and ligand concentrations (or more correctly, at different radiometal-to-ligand molar ratios) and monitored as a function of time in order to optimize the radiolabeling. As a continuation of this project, another full study on that particular topic is about to be submitted.

Minor:

  1. Although not the authors’ fault, there are a lot of formatting errors (e.g. page 3 line 105, 108; page 6 line 219; equation 11; page 8 line 264, 270, 280; and the t-rex (a favourite) in page 17 line 652) that forced a lot of guessing from the reviewer’s part.

The formatting errors do not appear in the word document. They correspond to greek letters. Nevertheless, we have corrected all the greek letters into symbol font. We do hope there will be anymore formatting issues.

  1. Page 2 line 61-62: Can the authors add a reference for the radical depolymerisation and oversulfation?

The reference has been added accordingly to the reviewer’s suggestion.

  1. Page 2 line 61: The authors use the same acronym for both the products and the processes to obtain them (i.e. EPS DR and EPS DRS). It might create some confusion. Would it be possible to differentiate the two things?

For sake of clarity, we have rephrased the sentences as “In order to promote its biological activities and to provide a GAG mimetic compound, the native EPS has been modified by first a radical depolymerization, then followed by an oversulfation step [12]. EPS-DR represents the native EPS that has undergone radical depolymerization, while EPS-DRS has undergone additional over-sulfation.”

  1. Page 2 figure 1: Na3OSO- groups on the molecules: it looks like there are 3 sodium atoms (Na3), not three oxygen atoms (OSO3).

We agree with the reviewer’s comment and the figure has been corrected

  1. Page 3 line 104: Scandium-44, with a half-life of 4h, is not compatible with the biological half-life of (full length) antibodies. The radiometal is more suitable for small proteins such as nanobodies and affibody molecules.

The reviewer’s comment is correct, a half-life of 4h is more suitable for small proteins such as nanobodies and affibody molecules. However, 44Sc has an isomeric state, 44mSc (T1/2 = 58.6 h), which can be co-produced with 44Sc and that has been proved to be considered it as an in-vivo PET generator 44mSc/44Sc (Nucl. Med. Biol. 2015, 42, 6, 544-529. doi: 10.1016/j.nucmedbio.2015.03.002; ibid 2014, 41, S36-43. doi :10.1016/j.nucmedbio.2013.11.004). 44mSc decays by internal transition (98.8%) to the ground state (44gSc) with an associated small secondary emission (Auger emission 2.74%).

  1. The authors should define the following acronyms/abbreviations: page 3 line 108, T1/2; Page 3 line 121, SEC; Page 9 line 298, CE-ICP-MS; Page 21, line 757: ICP-AES; Page 21, line 762: LQM.

A glossary has been added at the end of the manuscript.

  1. Page 3, line 117: For clarity, in this sentence can the authors indicate specifically the name of the molecules they are talking about instead of calling them “these molecules”?

Change has been done.

  1. Page 4, section from line 137 to line 163: This section might be more suitable for “Methods” than “Results”. And can it be stripped to its essential information?

It has been moved within the experimental section and shortened.

  1. Page 5, line 181: the unit is missing Lack of consistency: table 1 the unit is mL/g and in the text is mL g-1; page 6 lines 214 and 222.

The unit has been inserted and the format has been modified accordingly to the reviewer’s comment.

  1. AF4 is sometimes A4F;

The acronyms have been homogenized along the manuscript.

Figure 7, the graphs have different resolutions (some of them are difficult to read), inconsistent x-axis labels and legend/insert (a mix of decimal and base 10 numbers).

The Figure 7 is actually a mix of different figures / information. It is now Figure 5.

The units and axis are consistent with the parameter considered (i.e. ln hsp /c for the 3 polysaccharides, or for [h] for the 3 molecules)

For Fig 5 g and h, to evidence the influence of the electrolyte background, it is usually represented within a logarithmic scale.

  1. Page 6, Figure 3: Is there a reason why the authors show the fractogram of EPS-DRS only and not of EPS-DR?

The fractogram of EPS-DR is presented in Figure 4.

  1. Page 6, line 212: Are the authors sure of using equation 3 to calculate Rθ/K*c values?

It is corrected. Since a rearrangement has been done in the experimental section (see comment h), the number of the equations have been changed and in the revised version, it corresponds now to Eq. 3

Page 7 line 242: Again, the authors refer to the wrong equation. And Page 7, line 243: “see section 4.6” should be “see section 4.5”: It is now corrected

Page 12, line 428: It should be “a”, not “α”. It is corrected

  1. There is a little bit of inconsistency with the figures: figures 5 and 6 are missing. Page 12, line 410: the authors cite a non-existing figure 5.

The reviewer is right; there was a mistake in numbering the figures. The figures have been numbered correctly and the corresponding sections in the text they were quoted have been corrected as well.

  1. Page 8, line 273: “Higher than for heparin” not “higher for heparin”. The two sentences have very different meanings.

It has been changed.

  1. Page 11, line 375: The authors say: “Another limitation of SEC is the incapability of the technique to characterize high-molar mass polymers”. What mass is considered high? Can the authors indicate some values in the text?

The value has been added. Ultrahigh-molar-mass (M) polymers, generally considered those with M  ≥ 1 million g/mol (Anal. Bioanal. Chem. 394, 1887–1893 (2009). DOI:10.1007/s00216-009-2895-5)

  1. Page 13, line 468: The authors say: “This could be explained by considering the overall charges surrounding EPS-DR or -DRS chains that are more important compared to the heparin ones”. What do the authors mean with the term “important” referred to a charge? Do the authors think that this sentence is essential for the story?

This is of importance since there are more ionized functional groups on EPS-DR and -DRS than on heparin (which is a reference) and this has a consequence on the conformation of the polymers. That could then influence the complexation, as explained later in the manuscript.

  1. Page 14, line 521 to 537 and Page 15, line 573-581: Why do the authors focus so much on heparin? In the current state, those paragraphs look off-topic. If important for the story, can the authors find a way to connect them to the main subject (i.e. EPS)?

As highlighted in the introduction, “GAGs such as unfractionated heparin, and its low-molecular weight (LMW) derivatives, are generally used for prevention or treatment of venous thromboembolism that frequently occurs in cancer patients.” Heparin is thus a reference compound in our study. This is why when aiming to develop a “new” compound, it is very important to compare with a reference system.

  1. Page 17, section 4.3: The authors should fix some of the verb tenses. Line 641: what kind of filter was used (size, membrane, pore size)? what supplier?

It was a mistake; the samples were not filtered before the injection in the AF4 channel. This has been thus removed. Only the eluent was filtered through a 0.45µm filter of polyethersulfone (diameter 25 mm, Sigma Aldrich), but this corresponds to a “technical” detail on the way that AF4 is set from the provider, so it seems not very relevant to add this information in the experimental section..

  1. Page 21, lines 787 to 789: The authors talk about using the tools described in the manuscript to determine the consistency between different batches of polysaccharides. It is not clear if that was performed for this work or separately? Why not including those data in the manuscript?

In section 4.1, a sentence has been added for sake of clarity. “Several batches were produced in order to have sufficient amounts of EPS for performing the study. Each batch has been fully characterized using the methods described in the present paper.” 

Reviewer 2 Report

In the paper “Marine Exopolysaccharide Complexed with Scandium Aimed as Theranostic Agents”, Dr. Mazza and colleagues investigate the properties of expolysaccharides (EPS). The paper broadly investigate EPS compared to heparin, and specifically investigates the possibility of binding the element scandium (Sc). The binding of Sc would allow Sc-labelled EPS to be used in theranostics: Diagnostic for positron emission tomography (PET) with the beta+ emitter Sc-44, and therapeutically with the beta- emitter Sc-47. The study finds that EPS can be fractioned in fairly monodisperse fractions, suitable for therapeutic use, and that Sc-EPS has binding strength comparable to Sc-heparin.

The viewpoint of this reviewer is nuclear medicine, while several of the analysis methods described in the paper are new to me. As such, I have not been able to thoroughly evaluate the analysis. From a nuclear medicine perspective, the paper appears sound and relevant, and the specific points mentioned below are minor points. The paper is also well-written (although perhaps too lengthy), but as detailed below, the authors should perform an editorial review of their equation numbers, figure numbers and symbols.

SPECIFIC ISSUES

1. Please check equation numbers. The text in lines 155-157 refers to Eq. 1 and 2, but the form factor mentioned in line 156-157 does not appear to be part of any of these equations. Likewise in lines 211-212 where the reference to Equation 3 does not seem to fit the text, but might refer to Equation 4, and lines 241-242, where the references to equations 9 and 8 would fit better with equations 10 and 9. Is perhaps one equations missing after equation 2?

2. Please check non-Latin symbols, especially outside formulas. In lines 105 and 108 a strange symbol might mean “gamma”, and similar strange symbols are found in many other places (lines 155, 212, 219, 264, 270, 280, 456, and 652).

3. Please check figure numbers. All references to figures seem correct, but figures are numbered 1-4, 7-9, i.e. no figures are numbered 5 or 6.

4. Abbreviations: Not all abbreviations are defined on first use. In the abstract, MALS and FISRE are used without definition (while they are defined in the main text). SEC appears to be first mentioned in line 121 (SEC-MALS) without definition. Suggestion: For ease of the reader, perhaps a table of abbreviations may be added early in the paper?

5. In line 102, it is stated as an advantage that Sc-43 and Sc-44 decays to non-toxic Ca. While this is not wrong, it not especially relevant, due to the minute amounts daughter atoms. From the relation between activity (A), number of atoms (N), and half-life T, we get: N = A x T/ln(2). For for T = 14 000 s (about 4 h), and e.g. A = 10 GBq = 1E10 Bq, there will be about 2E14 atoms, or about 0.3 nmol. Thus, the total number of daughter atoms produced will be less than a nanomol. In so small amounts, it will generally to be unimportant what chemical element the daughter nucleus is. (A theoretical exception would be a case where a radioactive elute has decayed for so long that there are many orders of magnitude more of the daughter nuclei than remaining mother nuclei, and therefore many orders of magnitude times 0.3 nmol, but this appears unlikely to the case in practise.) It is suggested to omit the phrase “and decay to non-toxic Ca”. The 4 h half-life is relevant, yes.

6. In line 137, dn/dc is introduced. It is suggested to help the reader by defining n and c, e.g. “In dn/dc, n = RI (a notation often used in physics), while c = concentration.”

7. Blurry figures: Figure 1(a) and Figure 8 appear blurry (especially in my print-out). Has a lossy graphics format (e.g. jpg) been used? If possible, please use a non-lossy graphics format (e.g. tiff, png, gif, lossless jpg) for these images – as appears to be the case for all other figures, which are clear.

8. In headline of Table 1, it appears that a minus sign is missing in (SO4)2.

9. Lines 823-830: This is the draft text for Institutional Review Board Statement, not a statement for the paper. An IRB statement does not appear relevant in a study like the present, so it is suggested to simply omit this section, as opened for in the last line of the draft text.

Author Response

In the paper “Marine Exopolysaccharide Complexed with Scandium Aimed as Theranostic Agents”, Dr. Mazza and colleagues investigate the properties of expolysaccharides (EPS). The paper broadly investigate EPS compared to heparin, and specifically investigates the possibility of binding the element scandium (Sc). The binding of Sc would allow Sc-labelled EPS to be used in theranostics: Diagnostic for positron emission tomography (PET) with the beta+ emitter Sc-44, and therapeutically with the beta- emitter Sc-47. The study finds that EPS can be fractioned in fairly monodisperse fractions, suitable for therapeutic use, and that Sc-EPS has binding strength comparable to Sc-heparin.

The viewpoint of this reviewer is nuclear medicine, while several of the analysis methods described in the paper are new to me. As such, I have not been able to thoroughly evaluate the analysis. From a nuclear medicine perspective, the paper appears sound and relevant, and the specific points mentioned below are minor points. The paper is also well-written (although perhaps too lengthy), but as detailed below, the authors should perform an editorial review of their equation numbers, figure numbers and symbols.

The numbers of figures and equations have been corrected, and symbols have been checked.

SPECIFIC ISSUES

  1. Please check equation numbers. The text in lines 155-157 refers to Eq. 1 and 2, but the form factor mentioned in line 156-157 does not appear to be part of any of these equations. Likewise in lines 211-212 where the reference to Equation 3 does not seem to fit the text, but might refer to Equation 4, and lines 241-242, where the references to equations 9 and 8 would fit better with equations 10 and 9. Is perhaps one equations missing after equation 2?

The sections have been moved together in the experimental section now, leading thus to more consistency.

  1. Please check non-Latin symbols, especially outside formulas. In lines 105 and 108 a strange symbol might mean “gamma”, and similar strange symbols are found in many other places (lines 155, 212, 219, 264, 270, 280, 456, and 652).

See answer to the same comment from reviewer #1.

  1. Please check figure numbers. All references to figures seem correct, but figures are numbered 1-4, 7-9, i.e. no figures are numbered 5 or 6.

The reviewer is right and we do apologize for this. According to reviewer #1 comments as well, the numbers of figures and equations have been corrected, and symbols have been checked.

  1. Abbreviations: Not all abbreviations are defined on first use. In the abstract, MALS and FISRE are used without definition (while they are defined in the main text). SEC appears to be first mentioned in line 121 (SEC-MALS) without definition. Suggestion: For ease of the reader, perhaps a table of abbreviations may be added early in the paper?

A glossary has been added at the end of the manuscript. In addition, the abstract has been implemented with the full names of AF4-MALS and FISRE for sake of clarity. SEC has been fully named in line 121 together with its definition in the glossary section.

  1. In line 102, it is stated as an advantage that Sc-43 and Sc-44 decays to non-toxic Ca. While this is not wrong, it not especially relevant, due to the minute amounts daughter atoms. From the relation between activity (A), number of atoms (N), and half-life T, we get: N = A x T/ln(2). For for T = 14 000 s (about 4 h), and e.g. A = 10 GBq = 1E10 Bq, there will be about 2E14 atoms, or about 0.3 nmol. Thus, the total number of daughter atoms produced will be less than a nanomol. In so small amounts, it will generally to be unimportant what chemical element the daughter nucleus is. (A theoretical exception would be a case where a radioactive elute has decayed for so long that there are many orders of magnitude more of the daughter nuclei than remaining mother nuclei, and therefore many orders of magnitude times 0.3 nmol, but this appears unlikely to the case in practise.) It is suggested to omit the phrase “and decay to non-toxic Ca”. The 4 h half-life is relevant, yes.

The reviewer is absolutely right. Nonetheless, for radiopharmaceuticals compounds the regulations (FDA or Eur. Pharmacopeia) are very strict and the radionuclides of interest for Nuclear Medicine, with very few exceptions, must decay to a stable nucleus. For instance, if the decay product was a long-lived radionuclide, like Po-210, at the concentrations indicated by the reviewer (that are correct), they would be poisoning for the body. This is the reason why it has been indicated that the decay product is a stable element.

  1. In line 137, dn/dc is introduced. It is suggested to help the reader by defining n and c, e.g. “In dn/dc, n = RI (a notation often used in physics), while c = concentration.”

Thank you. According to the comments from reviewer #1, this section was too long and has been removed in the revised version of the manuscript.

  1. Blurry figures: Figure 1(a) and Figure 8 appear blurry (especially in my print-out). Has a lossy graphics format (e.g. jpg) been used? If possible, please use a non-lossy graphics format (e.g. tiff, png, gif, lossless jpg) for these images – as appears to be the case for all other figures, which are clear.

Figure 1a has been changed. Same for Figure 8. Hope it is now better in the printout.

  1. In headline of Table 1, it appears that a minus sign is missing in (SO4)2.

It has been added.

  1. Lines 823-830: This is the draft text for Institutional Review Board Statement, not a statement for the paper. An IRB statement does not appear relevant in a study like the present, so it is suggested to simply omit this section, as opened for in the last line of the draft text.

It has been removed accordingly.

Round 2

Reviewer 1 Report

The authors have carried out sufficient adjustments to the manuscript to grant acceptance.